# Membrane bridging by Munc13-1 is crucial for neurotransmitter release

Bradley Quade[1,2,3†], Marcial Camacho[4,5†], Xiaowei Zhao[1,6], Marta Orlando[4,5], Thorsten Trimbuch[4,5], Junjie Xu[1,2,3], Wei Li[1,2,3‡], Daniela Nicastro[1,6], Christian Rosenmund[4,5*], Josep Rizo[1,2,3*]

[1]Department of Biophysics, University of Texas Southwestern Medical Center, Dallas, United States; [2]Department of Biochemistry, University of Texas Southwestern Medical Center, Dallas, United States; [3]Department of Pharmacology, University of Texas Southwestern Medical Center, Dallas, United States; [4]Institut für Neurophysiologie, Charité - Universitätsmedizin, Berlin, Germany; [5]NeuroCure Cluster of Excellence, Berlin, Germany; [6]Department of Cell Biology, University of Texas Southwestern Medical Center, Dallas, United States

*For correspondence:
christian.rosenmund@charite.de (CR);
Jose.Rizo-Rey@UTSouthwestern.edu (JR)

†These authors contributed equally to this work

Present address: ‡National Laboratory of Biomacromolecules, Institute of Biophysics, Chinese Academy of Sciences, Beijing, China

Competing interests: The authors declare that no competing interests exist.

**Abstract** Munc13-1 plays a crucial role in neurotransmitter release. We recently proposed that the C-terminal region encompassing the $C_1$, $C_2B$, MUN and $C_2C$ domains of Munc13-1 ($C_1C_2BMUNC_2C$) bridges the synaptic vesicle and plasma membranes through interactions involving the $C_2C$ domain and the $C_1$-$C_2B$ region. However, the physiological relevance of this model has not been demonstrated. Here we show that $C_1C_2BMUNC_2C$ bridges membranes through opposite ends of its elongated structure. Mutations in putative membrane-binding sites of the $C_2C$ domain disrupt the ability of $C_1C_2BMUNC_2C$ to bridge liposomes and to mediate liposome fusion in vitro. These mutations lead to corresponding disruptive effects on synaptic vesicle docking, priming, and $Ca^{2+}$-triggered neurotransmitter release in mouse neurons. Remarkably, these effects include an almost complete abrogation of release by a single residue substitution in this 200 kDa protein. These results show that bridging the synaptic vesicle and plasma membranes is a central function of Munc13-1.

DOI: https://doi.org/10.7554/eLife.42806.001

## Introduction

The release of neurotransmitters by $Ca^{2+}$-triggered synaptic vesicle exocytosis is crucial for interneuronal communication. Exocytosis occurs in several steps that include tethering of synaptic vesicles to specialized sites of the presynaptic plasma membrane known as active zones, priming of the vesicles to a release-ready state(s) and $Ca^{2+}$-triggered fusion of the vesicles with the plasma membrane when an action potential causes $Ca^{2+}$ influx into the presynaptic terminal (*Südhof, 2013*). Release is exquisitely regulated by a sophisticated protein machinery that has been extensively characterized (*Brunger et al., 2018*; *Rizo, 2018*). Central components of this machinery are the SNAP receptors (SNAREs) synaptobrevin, syntaxin-1 and SNAP-25, which form a tight four-helix bundle called the SNARE complex that brings the vesicle and plasma membranes together and is critical for membrane fusion (*Hanson et al., 1997*; *Poirier et al., 1998*; *Söllner et al., 1993*; *Sutton et al., 1998*). The SNARE complex is disassembled by N-ethylmaleimide sensitive factor (NSF) and soluble NSF attachment proteins (SNAPs) (*Söllner et al., 1993*), whereas its assembly is orchestrated in an NSF-SNAP-resistant manner by Munc18-1 and Munc13s (*Ma et al., 2013*). The assembly pathway involves binding of Munc18-1 to a self-inhibited 'closed' conformation of syntaxin-1 (*Dulubova et al., 1999*; *Misura et al., 2000*) and to synaptobrevin to template SNARE complex formation (*Baker et al., 2015*; *Parisotto et al., 2014*; *Sitarska et al., 2017*) with the help of Munc13s, which facilitate

opening of syntaxin-1 to form the SNARE complex (*Ma et al., 2011*; *Richmond et al., 2001*; *Wang et al., 2017*; *Yang et al., 2015*). Synaptotagmin-1 acts as the major $Ca^{2+}$ sensor that triggers release through a combination of interactions with membranes and the SNARE complex (*Brewer et al., 2015*; *Chang et al., 2018*; *Fernández-Chacón et al., 2001*; *Zhou et al., 2015*).

Even with these and other advances, there are still fundamental questions that remain to be answered in order to understand the mechanisms of neurotransmitter release and its regulation. Particularly important is to elucidate the functions of mammalian Munc13s and their invertebrate homologues, Unc13s, because these large (ca. 200 kDa) proteins are essential for release (*Richmond et al., 1999*; *Varoqueaux et al., 2002*) and modulate exocytosis in multiple presynaptic plasticity processes through the various domains in its architecture (*Rizo and Südhof, 2012*). Munc13-1, the most abundant isoform in the mammalian brain, contains a variable N-terminal region with a $C_2A$ domain and a calmodulin-binding region (CaMb), as well as a conserved C-terminal region that includes the $C_1$, $C_2B$, MUN and $C_2C$ domains (*Figure 1A*). The $C_2A$ domain forms a homodimer and alternatively a heterodimer with the Rab3 effectors called RIMs (*Betz et al., 2001*; *Dulubova et al., 2005*; *Lu et al., 2006*), thus providing a switch that controls neurotransmitter release and couples exocytosis to diverse forms of Rab3- and RIM-dependent presynaptic plasticity (*Camacho et al., 2017*; *Deng et al., 2011*; *Rizo and Südhof, 2012*); the CaMb region mediates some forms of $Ca^{2+}$-dependent short-term plasticity (*Junge et al., 2004*); the $C_1$ domain mediates diacylglycerol (DAG)- and phorbol ester-dependent potentiation of release (*Basu et al., 2007*; *Rhee et al., 2002*); and the $C_2B$ domain acts as a $Ca^{2+}$- and $PIP_2$-dependent modulator of short-term plasticity (*Shin et al., 2010*). The MUN domain is a highly elongated module that is homologous to factors involved in tethering in diverse membrane compartments and is critical for opening syntaxin-1 (*Basu et al., 2005*; *Ma et al., 2011*; *Pei et al., 2009*; *Yu and Hughson, 2010*).

The Munc13 module that has remained more enigmatic is the $C_2C$ domain. Multiple evidence suggests that this domain is critical for Munc13 function (*Liu et al., 2016*; *Madison et al., 2005*; *Stevens et al., 2005*), but its biochemical properties and mechanism of action are not well understood. Based on sequence alignments, the Munc13-1 $C_2C$ domain is not predicted to bind $Ca^{2+}$ because it lacks some of the canonical aspartate residues that typically bind $Ca^{2+}$ in $C_2$ domains (*Rizo and Südhof, 1998*). Reconstitution studies of synaptic vesicle fusion and vesicle clustering assays suggested that the $C_2C$ domain binds to membranes, leading to a model whereby the conserved Munc13-1 C-terminal region bridges the synaptic vesicle and plasma membranes through respective interactions with the $C_2C$ domain and the $C_1$-$C_2B$ region on opposite ends of the MUN domain (*Liu et al., 2016*) (*Figure 1—figure supplement 1*). This model is consistent with the notion that the $C_1$ and $C_2B$ domains cooperate in binding to the plasma membrane through interactions with DAG and $PIP_2$, respectively (*Basu et al., 2007*; *Rhee et al., 2002*; *Shin et al., 2010*; *Xu et al., 2017*), and a role for Munc13-1 in bridging membranes seems natural given the homology of the MUN domain with tethering factors. However, no structure-function analysis of the $C_2C$ domain has been described, and the physiological relevance of the membrane bridging model has not been investigated.

The study presented here was designed to test this model and elucidate the function of the Munc13-1 $C_2C$ domain, which is critical to understand the mechanism of action of Munc13s. We show that the Munc13-1 C-terminal region can bridge two membranes through the ends of its elongated structure and that the $C_2C$ domain is essential for this ability. Moreover, impairment of the bridging activity by mutations in putative membrane-binding residues within the $C_2C$ domain correlates with disruption of synaptic vesicle docking, priming and neurotransmitter release. Our results show that, remarkably, a single point mutation in a 200 kDa protein such as Munc13-1 practically abolishes evoked neurotransmitter release, demonstrating the crucial importance of the membrane bridging activity for Munc13-1 function and for the sequence of events that lead to synaptic vesicle fusion.

## Results

### Functional consequences of deleting the Munc13-1 $C_2C$ domain

Using electrophysiological experiments in neuronal autaptic cultures, we previously showed that the conserved C-terminal region spanning the $C_1$, $C_2B$, MUN and $C_2C$ domains of Munc13-1

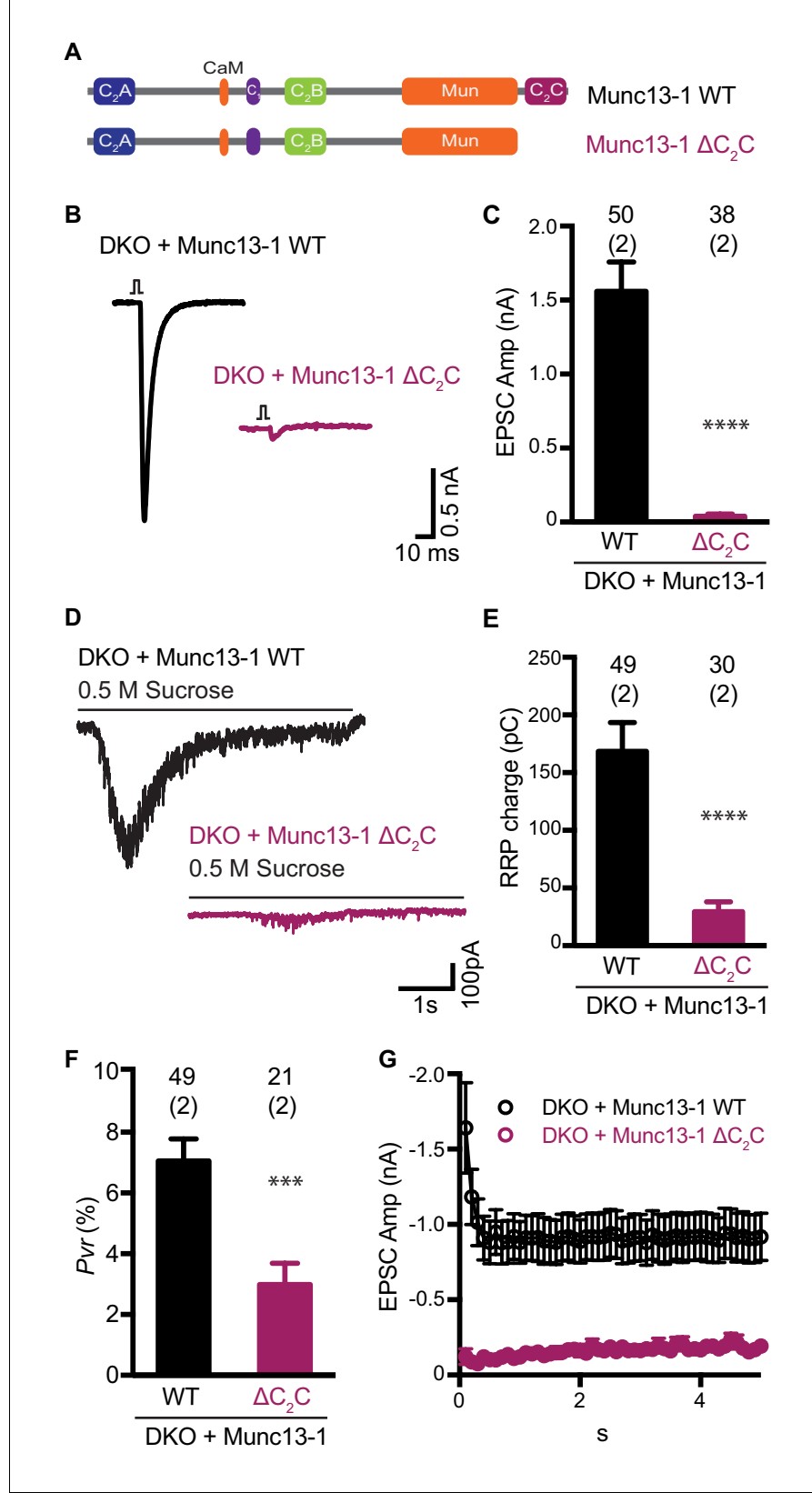

**Figure 1.** Functional effects caused by deleting the Munc13-1 $C_2C$ domain. (**A**) Cartoon depicting the domain structure of Munc13-1 and Munc13-1 $\Delta C_2C$. (**B**), Example EPSC traces recorded from *Munc13-1/2* DKO autaptic hippocampal neurons expressing either Munc13-1 WT (black) or Munc13-1 $\Delta C_2C$ (burgundy red). (**C**) Plot showing

*Figure 1 continued on next page*

*Figure 1 continued*

the average EPSC amplitudes obtained from the DKO neurons rescued with Munc13-1 WT or Munc13-1 $\Delta C_2C$. (D) Example traces of synaptic current responses induced by 5 s application of 500 mM sucrose from DKO neurons rescued with the WT and the $C_2C$ truncated mutant indicated above. (E) Plot of the average RRP charge for both groups. (F) Plot of the calculated *Pvr* in % for Munc13-1 WT and $C_2C$ truncated mutant. (G) Graph showing the absolute EPSC amplitudes in response to a train of 50 action potentials (APs) with an inter-stimulus interval (ISI) of 100 ms (10 Hz) for the WT and truncated $C_2C$ domain mutant. Numbers at the top of the bars represent the number of neurons pooled together for each group. Numbers in parentheses represent the number of cultures or replicates used. All data are mean ± SEM. Significance and p values were determined by Mann-Whitney test. ****p<0.0001: ***p<0.001.

DOI: https://doi.org/10.7554/eLife.42806.002

The following source data and figure supplement are available for figure 1:

**Source data 1.** Numerical description and statistics of data presented in *Figure 1*.

DOI: https://doi.org/10.7554/eLife.42806.004

**Figure supplement 1.** Model illustrating how the Munc13-1 C-terminal region can bridge the synaptic vesicle and plasma membranes.

DOI: https://doi.org/10.7554/eLife.42806.003

($C_1C_2BMUNC_2C$) can partially rescue the readily-releasable pool (RRP) and evoked neurotransmitter release in *Munc13-1/2* double knockout (DKO) neurons, while an analogous fragment lacking the $C_2C$ domain was practically unable to rescue release (*Liu et al., 2016*). These results supported the notion that the $C_2C$ domain is crucial for Munc13-1 function, but we later showed that the incomplete rescue obtained with $C_1C_2BMUNC_2C$ arises in part because removal of the N-terminal region containing the $C_2A$ domain impairs synaptic vesicle docking (*Camacho et al., 2017*). Since our model postulates that the $C_2C$ domain plays a key role in membrane bridging by the Munc13-1 C-terminal region and this mechanism might be at least partially redundant with the function of the $C_2A$ domain in docking, it became important to test the functional importance of the $C_2C$ domain in the context of full-length Munc13-1. For this purpose, we used a rescue approach with autaptic neuronal cultures from *Munc13-1/2* DKO mice, where $Ca^{2+}$-evoked release, spontaneous release and sucrose-induced release, which measures the readily release pool (RRP) of vesicles, are completely abolished (*Varoqueaux et al., 2002*).

Lentiviral expression of full-length wild type (WT) Munc13-1 in neuronal autaptic cultures from *Munc13-1/2* DKO mice robustly rescue evoked release, as observed previously (*Liu et al., 2016*), but almost no evoked release was observed when Munc13-1 lacking the $C_2C$ domain (Munc13-1 $\Delta C_2C$) was expressed (*Figure 1B,C*). Deletion of the $C_2C$ domain also reduced the RRP strongly, although the impairment was not as severe as that observed for evoked release (*Figure 1D,E*). As a result, the release probability of the few vesicles that were primed was decreased for the Munc13-1 $\Delta C_2C$ rescue compared with the WT rescue (*Figure 1F*). As expected from the decrease in vesicular release probability, we also found that synapses from neurons rescued with Munc13-1 $\Delta C_2C$ exhibited facilitation upon repetitive stimulation, unlike those rescued with WT Munc13-1 (*Figure 1G*). These results demonstrate that the Munc13-1 $C_2C$ domain plays a critical role in synaptic exocytosis, in agreement with previous results (*Liu et al., 2016*; *Madison et al., 2005*; *Stevens et al., 2005*), and show that this role is important for vesicle priming and also crucial for evoked neurotransmitter release.

## A Munc13-1 MUNC$_2$C fragment binds to membranes

Multiple attempts to express the isolated Munc13-1 $C_2C$ domain to characterize its structure and biochemical properties failed to yield soluble, properly folded protein fragments. However, a longer fragment including the $C_2C$ domain and the preceding MUN domain (MUNC$_2$C) can be readily expressed in bacteria (*Liu et al., 2016*), suggesting that the $C_2C$ domain requires packing against the MUN domain for proper folding. To confirm that the $C_2C$ domain is folded within the MUNC$_2$C fragment, we compared $^1H$-$^{13}C$ heteronuclear multiple quantum coherence (HMQC) spectra of per-deuterated samples of the Munc13-1 MUN domain and MUNC$_2$C fragment that were specifically $^1H$,$^{13}C$-labeled at Ile, Leu and Val methyl groups ($^2H$,$^{13}CH_3$-ILV-labeled). The spectrum of the MUNC$_2$C fragment contains additional cross-peaks in well-resolved regions (*Figure 2—figure*

*supplement 1A,B*), showing that the $C_2C$ domain is structured. In addition, the shifts observed in some of the cross-peaks of the MUN domain upon inclusion of the $C_2C$ domain support the notion that there are intramolecular interactions between the two domains.

Since yeast two-hybrid assays indicated that a C-terminal fragment spanning part of the MUN domain and the $C_2C$ domain of Munc13-1 bind to syntaxin-1 (*Betz et al., 1997*), we tested whether the $C_2C$ domain contributes to such binding using NMR spectroscopy. For this purpose, we acquired $^1H$-$^{15}N$ transverse relaxation optimized (TROSY) heteronuclear single quantum coherence (HSQC) spectra of $^{15}N$-labeled cytoplasmic region of syntaxin-1 (residues 2–253) in the absence and presence of unlabeled Munc13-1 MUN domain and $MUNC_2C$ fragment. Both fragments caused similar, limited broadening of the cross-peaks of syntaxin-1 (2–253) (*Figure 2—figure supplement 1C–E*), but all cross-peaks remained observable. Given the large size of these Munc13-1 fragments (residues 859–1516 and 859–1735, respectively), substantial binding would be expected to induce much stronger broadening (*Rizo et al., 2012*). Hence, these results show that the two fragments bind very weakly to the syntaxin-1 (2–253) fragment and that the $C_2C$ domain does not enhance the weak interaction involving the MUN domain, as the presence of the $C_2C$ domain in $MUNC_2C$ did not increase the broadening.

In previous experiments, we did not detect binding of the $MUNC_2C$ fragment to membranes in liposome co-floatation assays, but the $C_2C$ domain appeared to contribute to the ability of a fragment spanning the entire Munc13-1 C-terminal region ($C_1C_2BMUNC_2C$) to bridge liposomes containing synaptobrevin and a lipid composition resembling that of synaptic vesicles (V-liposomes) with liposomes containing syntaxin-1-SNAP-25 heterodimers and a lipid composition that mimics the plasma membrane (T-liposomes) (*Liu et al., 2016*). Hence, we hypothesized that the Munc13-1 $C_2C$ domain binds weakly to membranes and that such binding was not detectable in the co-floatation assays, but cooperativity between the $C_2C$ domains of two or more $C_1C_2BMUNC_2C$ molecules enables their liposome-liposome bridging activity. Note that membrane binding is the most common function of $C_2$ domains (*Rizo and Südhof, 1998*). Such binding is often mediated in a $Ca^{2+}$-dependent manner through loops that form $Ca^{2+}$-binding sites at the tip of a β-sandwich structure and these loops contain exposed basic and hydrophobic residues that can bind to negatively charged phospholipids and insert into membrane bilayers (*Chapman and Davis, 1998*; *Fernández-Chacón et al., 2001*; *Rizo and Südhof, 1998*). In addition, some $C_2$ domains contain a polybasic region on the side of the β-sandwich that can also contribute to membrane binding (e.g. the synaptotagmin-1 $C_2B$ domain [*Li et al., 2006*] and the RIM1 $C_2B$ domain [*de Jong et al., 2018*]). Although the Munc13-1 $C_2C$ domain is not expected to bind $Ca^{2+}$, it could bind lipids in a $Ca^{2+}$-independent manner through similar sequences. Indeed, models of the three-dimensional structure of the Munc13-1 $C_2C$ domain derived from its homology to $C_2$ domains of known structure such as the synaptotagmin-1 $C_2B$ domain (*Fernandez et al., 2001*) and the RIM1 $C_2B$ domain (*Guan et al., 2007*) consistently predicted that the $C_2C$ domain contains exposed basic and hydrophobic residues in its putative membrane-binding loops, as well as a polybasic region on the side of the β-sandwich. *Figure 2A* illustrates one of these models, highlighting the residues that we chose for mutagenesis in this study.

To test whether the $C_2C$ domain indeed binds to membranes, we used a GST-pulldown assay and designed a mutation that replaces an arginine and a phenylalanine from putative membrane-binding loops with glutamate (R1598E/F1658E). We immobilized WT and R1598E/F1658E mutant GST-$MUNC_2C$ fusion proteins on a GST-affinity resin, and checked that both samples had a comparable amount of protein (*Figure 2B*). The two samples, as well as a control sample of protein-free GST-affinity resin, were incubated with liposomes that had a lipid composition that resembles that of synaptic vesicles and 2% of a rhodamine-labeled lipid (rho-liposomes). We then centrifuged the samples and recorded emission fluorescence spectra of the flow-through. The spectra obtained from the eluate of the protein-free resin was analogous to that of the original lipid solution (not shown), indicating that the rho-liposomes were not retained by the resin. The fluorescence intensity observed in the spectra of the sample eluted from the R1598E/F1658E GST-$MUNC_2C$ resin was also very similar to that of the eluate from the protein-free resin whereas the sample eluted from the WT GST-$MUNC_2C$ resin had a much weaker intensity (*Figure 2C*). These results show that WT GST-$MUNC_2C$ efficiently bound to the rho-liposomes, while there was no detectable binding for the R1598E/F1658E GST-$MUNC_2C$ mutant. In analogous experiments where we loaded the resins with a four-fold larger amount of WT and R1598E/F1658E proteins, the R1598E/F1658E mutant again failed to

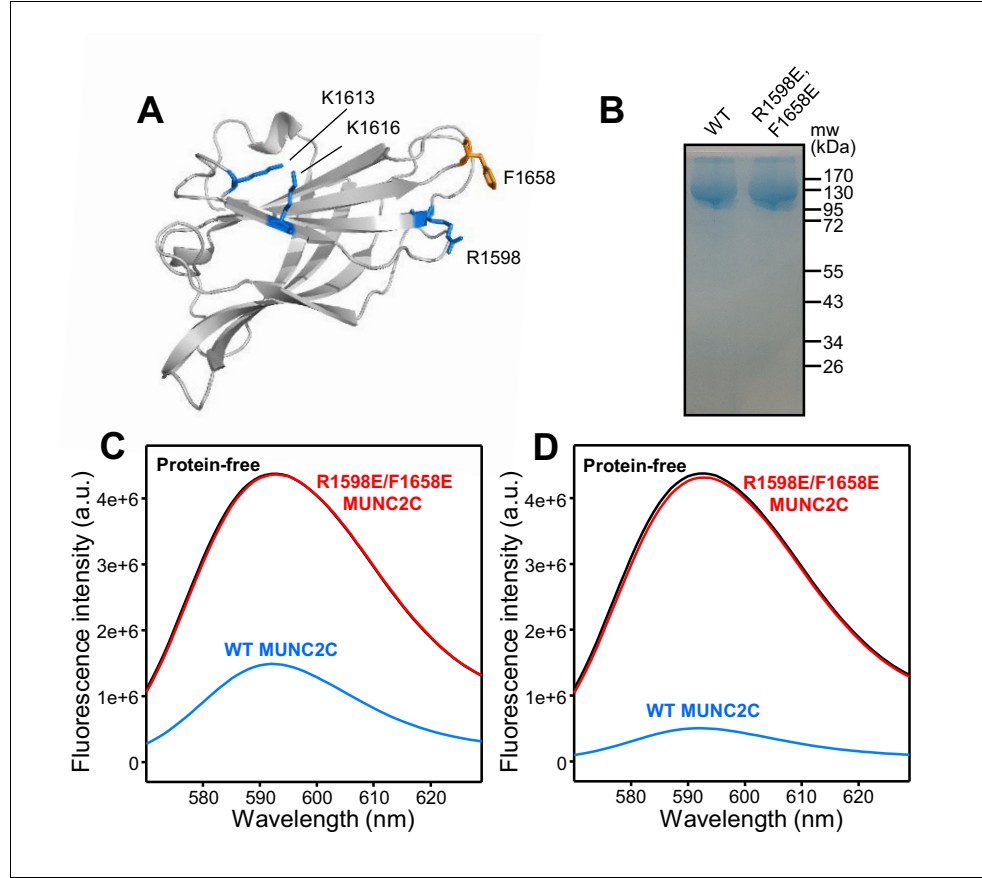

**Figure 2.** The MUNC$_2$C fragment binds to liposomes. (**A**) Ribbon diagram of a structural model of the Munc13-1 C$_2$C domain built based on the crystal structure of the RIM1 C$_2$B domain (*Guan et al., 2007*) (PDB accession code 2Q3X) and the sequence homology between the two C$_2$ domains. The side chains of the residues that were mutated in this study are shown as stick models. (**B**) Analysis of the resins used for GST-pulldown assays of liposome binding. Equal amounts of GST-affinity resins loaded with WT or R1598E/F1658E MUNC$_2$C were analyzed by SDS-PAGE and coomassie blue staining. (**C,D**) Fluorescence emission spectra showing that GST-affinity resins loaded with WT MUNC$_2$C bind to rho-liposomes and that the R1598E/F1658E mutation abolishes binding. GST-affinity resins that were protein free (black curves) or were loaded with WT MUNC$_2$C (blue curves) or R1598E/F1658E MUNC$_2$C (red curves) were incubated with rho-liposomes. The WT and mutant GST-MUNC$_2$C resins were loaded with four-fold larger amounts of proteins for the experiments of panel (**D**) than for those of panel (**C**). Samples were centrifuged and fluorescence emission spectra of the flow through were acquired. Note that comparable amounts of rho-liposomes were eluted from the protein-free and R1598E/F1658E GST-MUNC$_2$C resins, whereas much less rho-liposomes were eluted from the WT GST-MUNC$_2$C resins.

DOI: https://doi.org/10.7554/eLife.42806.005

The following figure supplement is available for figure 2:

**Figure supplement 1.** NMR analysis of the Munc13-1 MUNC$_2$C fragment and binding to syntaxin-1.

DOI: https://doi.org/10.7554/eLife.42806.006

bind rho-liposomes, whereas the WT protein retained most of the rholiposomes on the resin (*Figure 2D*), showing the specificity of the interaction. We note that the estimated amounts of proteins loaded on the resins were 2.3 and 9.2 nanomoles for the experiments of *Figure 2C and D*, respectively, while we used only 0.25 picomoles of liposomes (assuming a 100 nm diameter) for both sets of experiments. The fact that we observed robust liposome binding to GST-MUNC$_2$C in these assays but we did not observe binding of MUNC$_2$C to liposomes in co-floatation assays (*Liu et al., 2016*) suggests that each rho-liposome binds to multiple GST-MUNC$_2$C molecules, which by virtue of their attachment to the resin can cooperate in retaining the rho-liposomes. Overall, these results

show that the Munc13-1 MUNC$_2$C fragment indeed binds to membranes, and that this activity is abolished by the R1598E/F1658E mutation.

## The C$_2$C domain is required for membrane bridging by the Munc13-1 C-terminal region

The MUN domain has a highly elongated structure (*Xu et al., 2017*; *Yang et al., 2015*), with the C$_1$-C$_2$B region and the C$_2$C domain attached at opposite ends. Since the C$_1$ and C$_2$B domains bind to DAG and PIP$_2$, respectively, the observation that C$_1$C$_2$BMUNC$_2$C bridges T- and V-liposomes suggested that this activity involves interactions of the C$_1$-C$_2$B region with the T-liposomes and the C$_2$C domain with the V-liposomes (*Figure 1—figure supplement 1*). To test this model and directly visualize whether the C$_1$C$_2$BMUNC$_2$C fragment can indeed bridge two membranes through the ends of its highly elongated structure, we acquired cryo-electron tomography (cryo-ET) images of reconstitution reactions where T-liposomes and V-liposomes were mixed together with Munc13-1 C$_1$C$_2$BMUNC$_2$C, Munc18-1, NSF and αSNAP. Indeed, we observed many instances where two liposomes were bridged by highly elongated densities (*Figure 3A–E*). Measurements made for 70 of these highly elongated densities yielded an average length of 22 nm, consistent with the approximate length that can be predicted for C$_1$C$_2$BMUNC$_2$C based on the crystal structure of the Munc13-1 C$_1$C$_2$BMUN fragment (ca. 20 nm long [*Xu et al., 2017*]). Because the three-dimensional structures of all the other proteins included in the samples are known and none of them has such an elongated shape (*Rizo, 2018*), these densities can be attributed unambiguously to the Munc13-1 C$_1$C$_2$BMUNC$_2$C fragment. We note that liposomes generally formed clusters where each liposome pair was bridged by at least one, and often more, C$_1$C$_2$BMUNC$_2$C molecules that likely cooperate in clustering. In this context, it is worth noting that super-resolution imaging revealed the formation of supramolecular assemblies by multiple Munc13-1 molecules at presynaptic release sites (*Sakamoto et al., 2018*).

Disruption of binding to the C$_2$C domain is expected to impair the ability of C$_1$C$_2$BMUNC$_2$C to bridge liposomes but to leave the C$_1$-C$_2$B region unaffected, thus allowing binding of C$_1$C$_2$BMUNC$_2$C to liposomes through one end of the molecule. To directly visualize this prediction for the C$_1$C$_2$BMUNC$_2$C R1598E/F1658E mutant, we again used cryo-ET and the same liposome preparations used for WT C$_1$C$_2$BMUNC$_2$C. The liposomes generally appeared more disperse in specimens containing the C$_1$C$_2$BMUNC$_2$C R1598E/F1658E mutant (*Figure 3F–J*) than those containing WT C$_1$C$_2$BMUNC$_2$C (*Figure 3A–E*). It was more difficult to identify C$_1$C$_2$BMUNC$_2$C molecules for the R1598E/F1658E mutant than for the WT protein, which we attribute to the fact that the R1598E/F1658E mutation disrupts its membrane-bridging activity and the protein may then have a higher chance to be sequestered at the water-air interface. Nevertheless, we were able to identify 42 C$_1$C$_2$BMUNC$_2$C R1598E/F1658E mutant molecules, and all of them were bound to a single liposome. In contrast, among 123 molecules of WT C$_1$C$_2$BMUNC$_2$C that we identified, 78 were bridging two liposomes and 45 were bound to a single liposome (25 among these 45 likely did not bridge liposomes due to steric hindrance caused by other C$_1$C$_2$BMUNC$_2$C molecules at the liposome-liposome interface). These results are consistent with dynamic light scattering (DLS) data showing complete abrogation of liposome clustering by the R1598E/F1658E mutation (see below) and strongly support the notion that binding of the C$_2$C domain to lipids is key for the membrane-bridging activity of C$_1$C$_2$BMUNC$_2$C.

The cryo-ET images provide a direct visualization of how the Munc13-1 C$_1$C$_2$BMUNC$_2$C fragment can bridge two membranes through sequences located at opposite ends of the MUN domain, as previously proposed based on DLS experiments that revealed the ability of this fragment to cluster V- and T-liposomes (*Liu et al., 2016*). To ensure that the bridging activity indeed involves interactions of C$_1$C$_2$BMUNC$_2$C with the membranes and does not depend on binding to proteins, we performed clustering assays monitored by DLS using mixtures of protein-free liposomes with the same lipid compositions as V- and T-liposomes (referred to as SV-liposomes and PM-liposomes because these lipid compositions mimic those of synaptic vesicles and the plasma membrane, respectively). The data showed that C$_1$C$_2$BMUNC$_2$C robustly clusters SV- and PM-liposomes in the absence of Ca$^{2+}$ and that Ca$^{2+}$ does not substantially increase this activity (*Figure 4A*), as observed previously with V- and T-liposome mixtures (*Liu et al., 2016*). These results demonstrate that membrane bridging involves direct interactions of C$_1$C$_2$BMUNC$_2$C with the two apposed membranes.

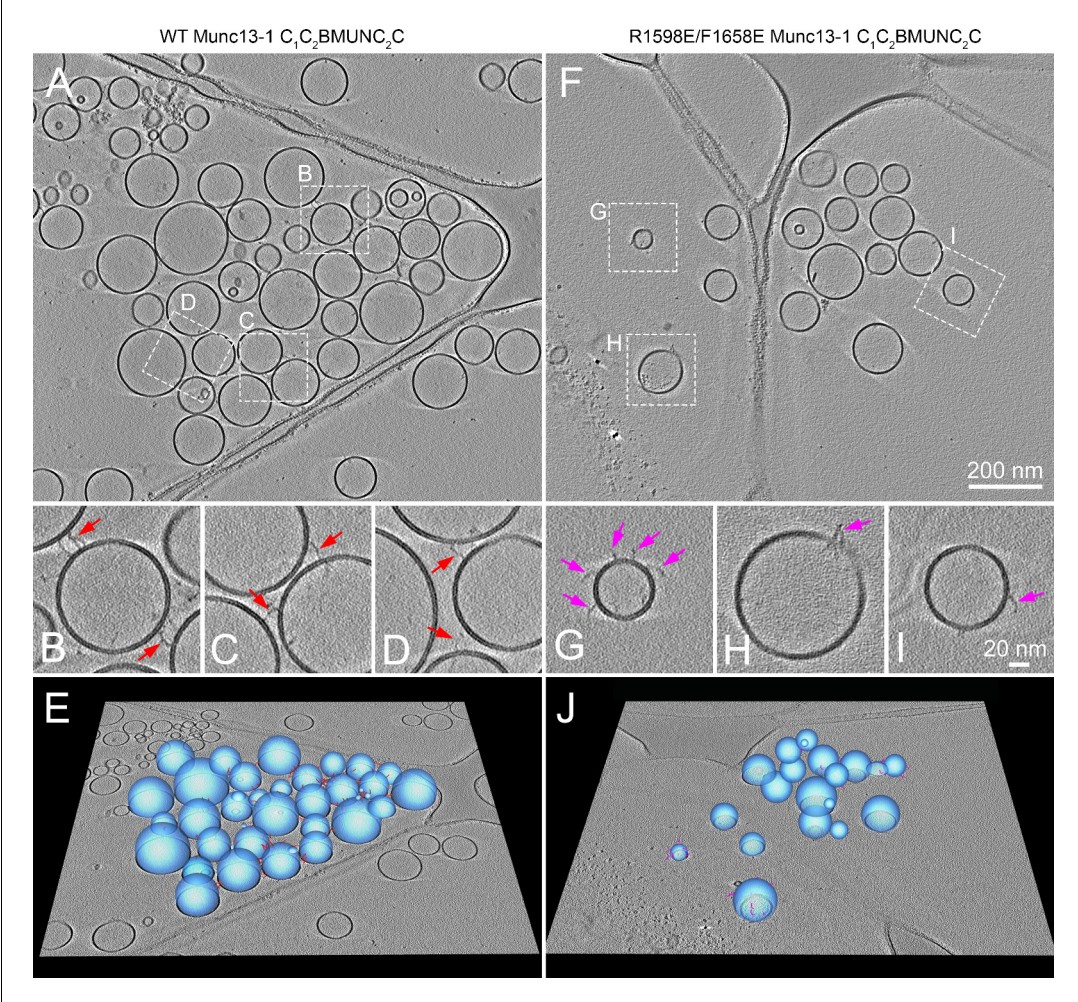

**Figure 3.** Cryo-ET reconstructions showing that Munc13-1 $C_1C_2BMUNC_2C$ can bridge two membranes. Specimens were prepared following our standard protocol to analyze lipid and content mixing between V- and T-liposomes in the presence of Munc18-1, NSF, αSNAP, 0.1 μM EGTA, and either WT (**A–E**) or R1598E/F1658E mutant (**F–J**) Munc13-1 $C_1C_2BMUNC_2C$ (see Materials and methods). (**A,F**) Tomographic slices provide an overview of the reaction mixtures including WT (**A**) or R1598E/F1658E mutant (**F**) Munc13-1 $C_1C_2BMUNC_2C$. (**B–D, G–I**) Zoom-in of the regions outlined in (**A**) and (**F**), respectively. However, note that the tomographic slices may vary slightly in z-height to optimize the visualization of the elongated densities corresponding to Munc13-1 $C_1C_2BMUNC_2C$. The majority of elongated densities of the WT protein bridge two liposomes (red arrows in **B–D**), whereas the elongated densities of the R1598E/F1658E mutant protein are bound to a single liposome (pink arrows in **G–I**). (**E,J**) 3D graphical models of the tomographic reconstructions show the vesicles (blue) and elongated densities of WT (red) and R1598E/F1658E mutant (pink) Munc13-1 $C_1C_2BMUNC_2C$ in 3D.

DOI: https://doi.org/10.7554/eLife.42806.007

The importance of the $C_2C$ domain for bridging might be questioned because the $C_1C_2BMUN$ fragment that we used in previous studies was also able to cluster liposomes (*Liu et al., 2016*). However, in this previous study , we noted that the $C_1C_2BMUN$ fragment used ended at residue 1531 and that the sequence spanning residues 1517 to 1531 is not part of the folded structure of the MUN domain. This sequence is highly hydrophobic and is probably folded in the $C_1C_2BMUNC_2C$ fragment, but is not observable in the structure of $C_1C_2BMUN$ (*Xu et al., 2017*). Hence, this sequence is exposed and likely mediates non-specific binding to membranes, which explains the ability of the $C_1C_2BMUN$ fragment to cluster liposomes (*Liu et al., 2016*). Therefore, to test to what extent the $C_2C$ domain is important for the vesicle clustering ability of the Munc13-1 C-terminal region, we prepared a new $C_1C_2BMUN$ fragment that ends at residue 1516 ($C_1C_2BMUN1516$) and hence lacks the C-terminal hydrophobic sequence. DLS assays showed that, in contrast to $C_1C_2BMUNC_2C$, the $C_1C_2BMUN1516$ fragment exhibited no clustering ability (*Figure 4A*). These

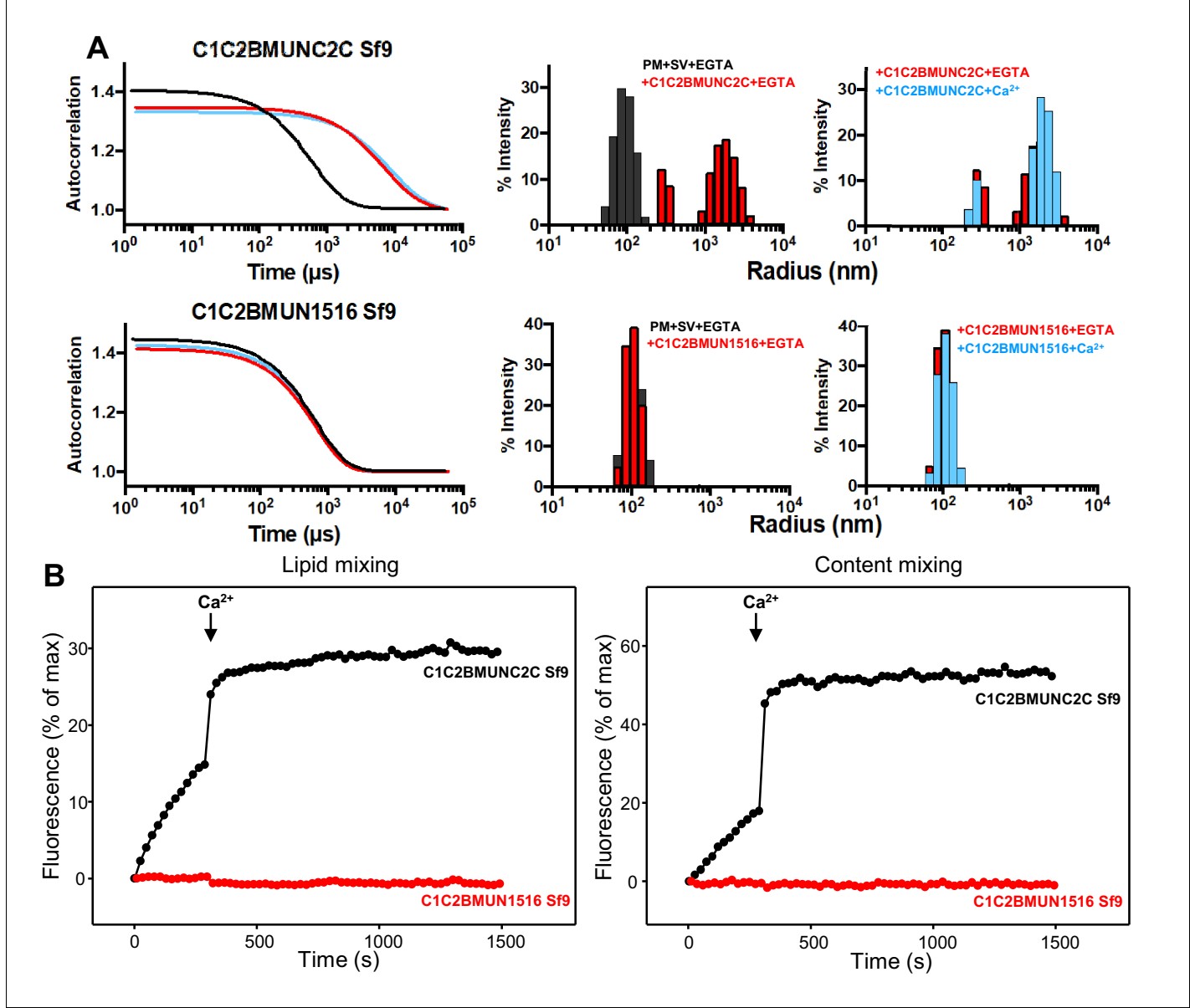

**Figure 4.** The Munc13-1 $C_2C$ domain is required for membrane bridging by the Munc13-1 C-terminal region. (A) DLS analysis of the ability of the Munc13-1 $C_1C_2BMUNC_2C$ or $C_1C_2BMUN1516$ fragments expressed in Sf9 cells to cluster SV- and PM-liposomes. The diagrams on the left show the autocorrelation curves observed for a mixture of the SV- and PM-liposomes alone in the presence of EGTA (black curves), or SV- and PM-liposomes together with the indicated Munc13-1 fragment in the presence of 0.1 mM EGTA (red curves) or 0.5 mM $Ca^{2+}$ (blue curves). The diagrams on the right show the particle size distributions corresponding to these curves, with the same color coding. (B) The $C_2C$ domain is required for the ability of the Munc13-1 C-terminal region to support liposome fusion in a reconstituted assay. Lipid mixing (left) between V- and T-liposomes was measured from the fluorescence de-quenching of Marina Blue-labeled lipids and content mixing (right) was monitored from the development of FRET between PhycoE-Biotin trapped in the T-liposomes and Cy5-Streptavidin trapped in the V-liposomes. The assays were performed in the presence of Munc18-1, NSF, αSNAP and the indicated Munc13-1 fragments. Experiments were started in the presence of 100 µM EGTA and 5 mM streptavidin, and $Ca^{2+}$ (600 µM) was added after 300 s.
DOI: https://doi.org/10.7554/eLife.42806.008

The following figure supplements are available for figure 4:

**Figure supplement 1.** $C_1C_2BMUNC_2C$ and $C_1C_2BMUN1516$ expressed in E. coli behave similarly to the same proteins expressed in Sf9 cells.
DOI: https://doi.org/10.7554/eLife.42806.009

**Figure supplement 2.** The $C_1C_2BMUN1516$ fragments bind to PM-liposomes.
DOI: https://doi.org/10.7554/eLife.42806.010

results strongly support the notion that the $C_2C$ domain is indeed crucial for the membrane bridging activity of the Munc13-1 C-terminal region.

All our previous studies with large Munc13-1 fragments used proteins expressed in Sf9 insect cells. As bacterial expression of the Munc13-1 $C_1C_2BMUN$ fragment ending at 1531 was recently described (*Kreutzberger et al., 2017*), we prepared new vectors for expression of Munc13-1 $C_1C_2BMUNC_2C$ and $C_1C_2BMUN1516$ in *E. coli*. Although the expression yields of both new fragments were modest, they were sufficient to obtain milligram quantities. DLS experiments showed that the bacterially expressed $C_1C_2BMUNC_2C$ and $C_1C_2BMUN1516$ fragments have analogous ability, or lack thereof, to cluster SV- and PM-liposomes as the corresponding fragments expressed in Sf9 cells (*Figure 4—figure supplement 1A*). We also measured the ability of these fragments to support fusion between reconstituted V- and T-liposomes in the presence of Munc18-1, NSF and αSNAP using an established assay that simultaneously measures lipid and content mixing (*Liu et al., 2016*). The $C_1C_2BMUNC_2C$ fragments expressed in Sf9 insect cells and *E. coli* exhibited comparable activities, with slow lipid and content mixing in the absence of $Ca^{2+}$ and fast fusion upon $Ca^{2+}$ influx (*Figure 4B* and *Figure 4—figure supplement 1B*). In contrast, the $C_1C_2BMUN1516$ fragments expressed in Sf9 insect cells and *E. coli* were both inactive, which correlates with the vesicle clustering results and shows the critical importance of the $C_2C$ domain for Munc13-1 to support fusion in these assays. We also note that the $C_1C_2BMUN1516$ fragments made in Sf9 insect cells and in bacteria exhibited the expected chromatographic behavior in gel filtration, with elution volume a little larger than the $C_1C_2BMUNC_2C$ fragments (16.5 versus 16.2 ml in a Superdex S200 10/300 GL column), and a similar ability to bind to PM-liposomes in co-sedimentation assays as $C_1C_2BMUNC_2C$ fragments (*Figure 4—figure supplement 2*). These results indicate that the $C_1C_2BMUN1516$ fragments are properly folded and retain the ability of the $C_1$-$C_2B$ region to bind to liposomes containing DAG and $PIP_2$, but cannot bridge these liposomes to SV-liposomes because they lack the $C_2C$ domain.

## Membrane bridging by $C_1C_2BMUNC_2C$ is crucial for its ability to support liposome fusion

The R1598E/F1658E mutation, which disrupts binding of $MUNC_2C$ to membranes (*Figure 2*), provides a useful tool to probe the functional importance of the membrane bridging activity of the Munc13-1 $C_1C_2BMUNC_2C$ fragment, but multiple mutations are ideally required to establish clear correlations between the bridging activity and Munc13-1 function. Thus, we prepared bacterially expressed versions of Munc13-1 $C_1C_2BMUNC_2C$ that contained the double R1598E/F1658E mutation, single R1598E and F1658E mutations, which might have milder effects, and a double residue substitution (K1613A/K1616A) in the polybasic region that may also participate in membrane binding (see *Figure 2A*).

DLS assays that monitored clustering between SV- and PM-liposomes revealed that the K1613A/K1616A mutation partially disrupts the clustering activity of $C_1C_2BMUNC_2C$ (*Figure 5*). The single R1598E and F1658E mutations disrupted vesicle clustering strongly, although the R1598E mutant appeared to retain a slight clustering ability. Clustering was completely abolished by the R1598E/F1658E mutation. These results demonstrate the critical importance of the $C_2C$ domain loops for membrane bridging by the Munc13-1 $C_1C_2BMUNC_2C$ fragment, and show that the $C_2C$ domain polybasic region also contributes to this activity. We note that, in principle, the hydrophobic sequence spanning residues 1517 to 1531 might be responsible for lipid binding, and the effects of the mutations in the $C_2C$ domain could arise from long-range effects due to changes in the overall electrostatic potential that increase the repulsion with the membranes. However, the fact that the $C_2C$ domain cannot be expressed in soluble form in isolation while soluble $MUNC_2C$ is readily expressed suggests that the hydrophobic sequence spanning residues 1517 to 1531 is folded and forms part of the interface between the MUN and $C_2C$ domains in fragments that contain both domains. Moreover, it is unlikely that long-range effects due to changes in overall electrostatic potential can explain the dramatic disruption of liposome binding (*Figure 2*) and clustering (*Figure 5*) caused by mutations in the putative membrane-binding loops of the $C_2C$ domain. In addition, the K1613A/K1616A mutation removes two positive charges and has a moderate effect on clustering, whereas the F1658E mutation has a very strong effect on clustering while introducing only one negative charge. Conversely, the effects of the mutations can be readily rationalized by the accumulated knowledge on membrane binding to $C_2$ domains, which predicts that F1658 is a key residue that inserts into the

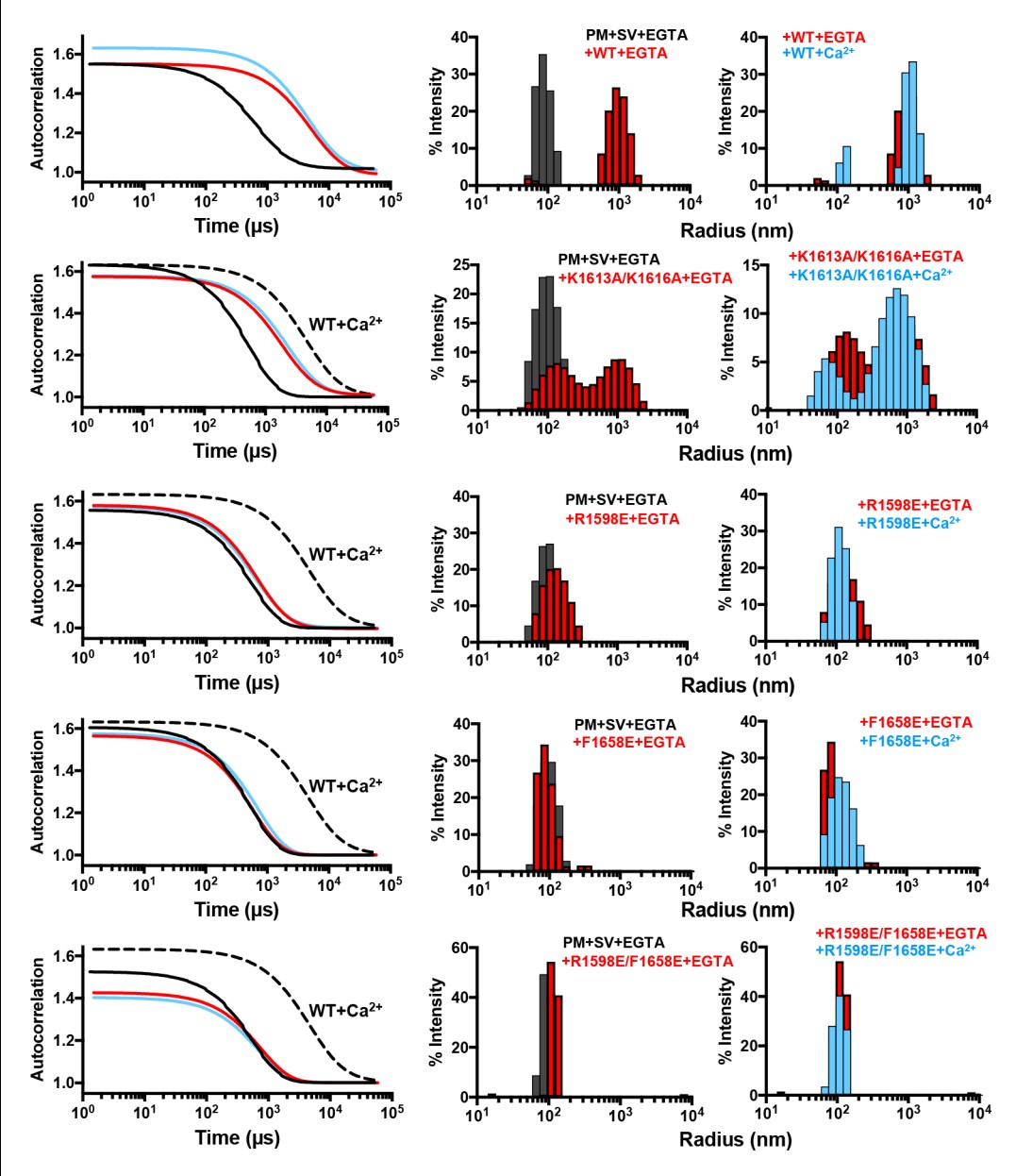

**Figure 5.** Mutations in putative membrane-binding sites of the Munc13-1 $C_2C$ domain disrupt the membrane-bridging activity of the Munc13-1 C-terminal region. DLS analysis of the ability of WT and mutant Munc13-1 $C_1C_2BMUNC_2C$ fragments to cluster SV- and PM-liposomes. The plots on the left show the autocorrelation curves observed for a mixture of the V- and T-liposomes alone in the presence of EGTA (black curves), or SV- and PM-liposomes together with the indicated Munc13-1 fragment in the presence of 0.1 mM EGTA (red curves) or 0.5 mM $Ca^{2+}$ (blue curves). In the plots corresponding to the mutants, the data obtained with WT $C_1C_2BMUNC_2C$ in the presence of $Ca^{2+}$ are shown by the dashed black curves for comparison. The diagrams on the right show the particle size distributions corresponding to these curves, with the same color coding.
DOI: https://doi.org/10.7554/eLife.42806.011

bilayer to mediate binding (*Chapman and Davis, 1998*; *Rhee et al., 2005*), while the polybasic region including K1613 and K1616 is expected to contribute to membrane binding but to a more moderate extent (*Li et al., 2006*).

To investigate the impact of these mutations on the ability of Munc13-1 $C_1C_2BMUNC_2C$ to support membrane fusion in vitro, we monitored lipid and content mixing between reconstituted V- and

T-liposomes in the presence of Munc18-1, NSF and αSNAP. In initial experiments, we used $C_1C_2BMUNC_2C$ fragments at 0.1 µM concentration, which allows better discrimination of the effects of mutations than the standard concentrations we normally used in these assays (0.5 µM) (*Xu et al., 2017*) and somewhat decreases the activity of the WT $C_1C_2BMUNC_2C$ fragment (*Figure 6—figure supplement 1A*). At this concentration, the K1613A/K1616A mutation considerably impaired fusion, whereas the R1598E, F1658E and R1598E/F1658E mutations completely abolished fusion (*Figure 6A*). To better characterize the effects of the mutations, we then performed titrations where the mutant $C_1C_2BMUNC_2C$ fragments were added at different concentrations. The K1613A/K1616A mutant was much more active at 0.25 and 0.5 µM concentrations than at 0.1 µM, whereas at 0.75 µM K1613A/K1616A we observed a slightly decreased activity that may arise because of appreciable precipitation (*Figure 6B*). The R1598E was able to support a small amount of lipid mixing at 0.5–2.5 µM concentrations, whereas the F1658E and R1598E/F1658E supported only very small amounts of lipid mixing at 2.5 µM concentration, and any content mixing supported by these three mutants was close to the noise level (*Figure 6B*). These observations were reproduced in multiple experiments with different liposome preparations and were confirmed by quantification of the amounts of lipid and content mixing observed after 500 s of reaction with 0.1 µM WT and K1613A/K1616A mutant, and of the lipid mixing observed after 1000 s for 0.5 µM WT and R1598E, F1658E and R1598E/F1658E mutants (*Figure 6—figure supplement 1B,C*). Overall, these results show that the F1658E and R1598E/F1658E mutations almost completely abolish the ability of Munc13-1 $C_1C_2BMUNC_2C$ to support membrane fusion, whereas the R1598E mutation causes a strong disruption, and the K1613A/K1616A mutation induces only a moderate impairment, mirroring the liposome clustering data.

## The mutations in the $C_2C$ domain disrupt synaptic vesicle docking, priming and release

To examine the functional consequences of the mutations in the Munc13-1 $C_2C$ domain, we turned again to rescue experiments in neuronal autaptic cultures from *Munc13-1/2* DKO mice and compared the release observed upon lentiviral expression of full-length Munc13-1 bearing mutations in the $C_2C$ domain with those observed with the WT rescue. The R1598E, F1658E and R1598E/F1658E mutations severely impaired spontaneous, evoked and sucrose-induced release, and the effects were particularly strong for evoked release, which was almost abolished by the F1658E and R1598E/F1658E mutations (*Figure 7A–F*). As a consequence of the stronger impairment of evoked release compared to sucrose-induced release, the three mutations led to decreases in the vesicular release probability (*Figure 7G*), as observed for the Munc13-1 ΔC₂C mutant (*Figure 1F*). Correspondingly, the paired-pulsed ratios measured for the three mutants were larger than that observed for WT Munc13-1 (*Figure 7H*), and all the mutant rescues exhibited facilitation upon repetitive stimulation, in contrast to the slight depression observed in the WT rescue (*Figure 7I*). The WT and mutant Munc13-1 proteins all exhibited presynaptic localization and were expressed at comparable levels (*Figure 7—figure supplement 1*), showing that the differences in electrophysiological parameters do not arise from mislocalization or aberrant overexpression. In a separate set of experiments, we analyzed the functional effects of the K1613A/K1616A mutation, using WT Munc13-1 again as positive control. This mutation did not impair spontaneous release but led to a moderate decrease in evoked release, and also appeared to decrease the RRP but the difference to WT was not statistically significant (*Figure 8A–F*). There was also no significant difference in the vesicular release probability and the paired-pulse ratios measured for rescue with WT and K1613A/K1616A mutant Munc13-1 (*Figure 8G,H*), although the K1613A/K1616A mutant displayed a milder depression upon repetitive stimulation than WT Munc13-1 (*Figure 8I*).

These results show that the ability of these various mutations in the Munc13-1 $C_2C$ domain to impair liposome clustering and fusion in vitro correlates well with the functional effects of these mutations on synaptic vesicle priming and $Ca^{2+}$-triggered neurotransmitter release in neurons. We also tested whether overexpression of the four Munc13-1 mutants in neurons from WT mice yielded differences in spontaneous, evoked and sucrose-induced release with respect to overexpression of WT Munc13-1, but we did not observe any significant differences that would suggest a dominant negative effect of the mutant fragments (*Figure 7—figure supplement 2*).

Previous studies that used high-pressure freezing/freeze substitution of organotypic hippocampal slice cultures and electron tomography showed that synaptic vesicle docking is strongly impaired in

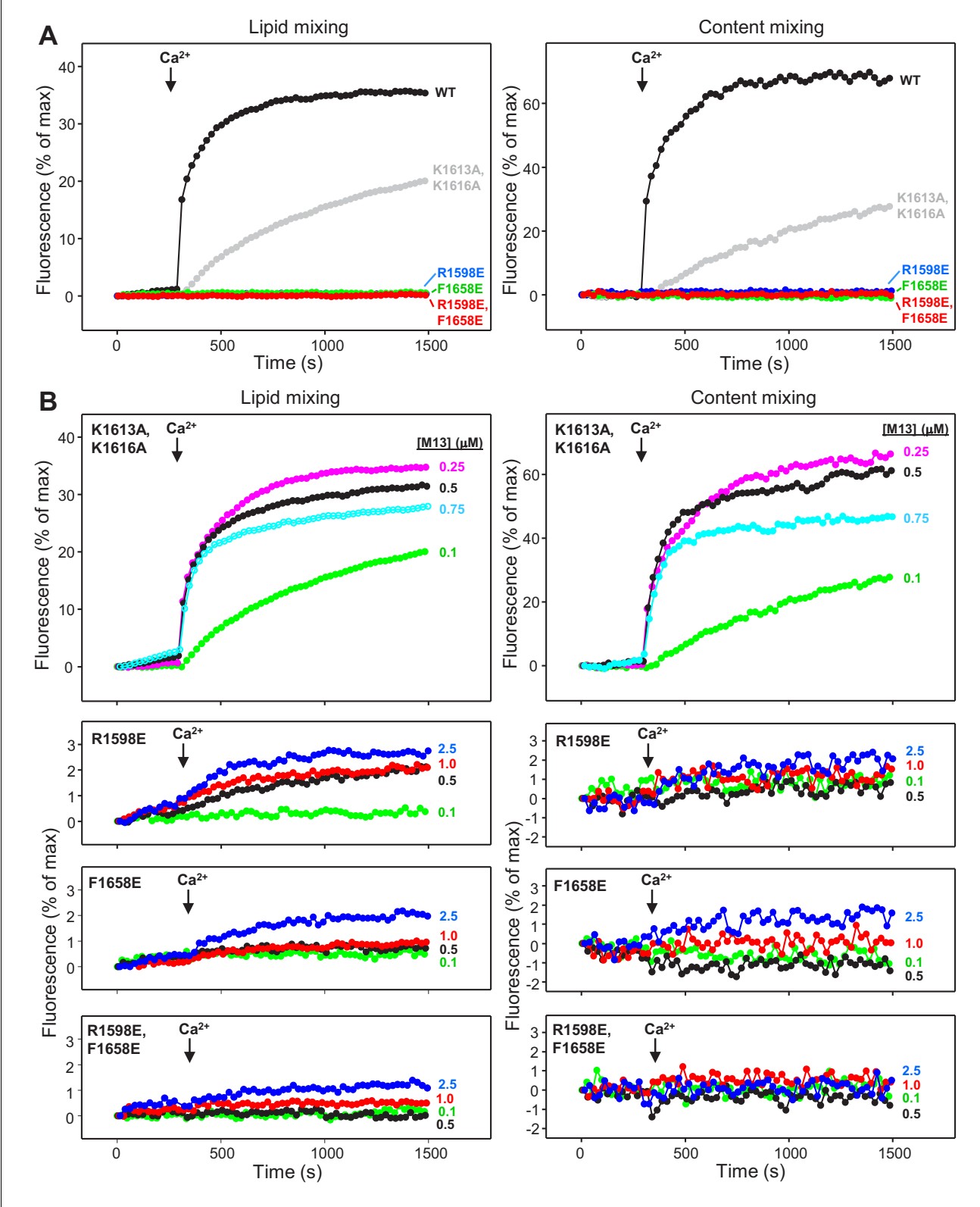

**Figure 6.** Mutations in putative membrane-binding sites of the Munc13-1 C$_2$C domain disrupt the ability of the Munc13-1 C-terminal region to support liposome fusion in a reconstituted assay. (**A**) Lipid mixing (left) between V- and T-liposomes was measured from the fluorescence de-quenching of Marina Blue-labeled lipids and content mixing (right) was monitored from the development of FRET between PhycoE-Biotin trapped in the T-liposomes and Cy5-Streptavidin trapped in the V-liposomes. The assays were performed in the presence of Munc18-1, NSF, αSNAP and 0.1 µM concentrations of

*Figure 6 continued on next page*

*Figure 6 continued*

WT or mutant Munc13-1 fragments, as indicated by the color codes. Experiments were started in the presence of 100 µM EGTA and 5 mM streptavidin, and Ca$^{2+}$ (600 µM) was added after 300 s. (B) Analogous lipid and content mixing assays performed with different concentrations of mutant Munc13-1 $C_1C_2$BMUNC$_2$C fragments as indicated. Note that the scale of the y-axis was expanded in the lower plots to help to visualize the small amounts of lipid and content mixing observed.

DOI: https://doi.org/10.7554/eLife.42806.012

The following figure supplement is available for figure 6:

**Figure supplement 1.** Mutations in putative membrane-binding sites of the Munc13-1 $C_2$C domain disrupt the ability of the Munc13-1 C-terminal region to support liposome fusion in a reconstituted assay.

DOI: https://doi.org/10.7554/eLife.42806.013

*Munc13-1/2* DKO neurons, defining docking as vesicles that appear to be in direct contact with pre-synaptic active zone membranes (*Imig et al., 2014*). Here, we used an analogous approach to study the impact of the four Munc13-1 point mutations on the ability of Munc13-1 to support synaptic vesicle docking. In this analysis, we also included the Munc13-1 $\Delta C_2$C mutant. Significant defects in docking were observed for all Munc13-1 mutants, with the R1598E/F1658E mutation having the strongest effect and the K1613A/K161A mutation the mildest (*Figure 9A,B*). A plot of the normalized number of docked synaptic vesicles observed for the WT and mutant Munc13-1 fragments against the RRP charge shows a strong correlation (*Figure 9C*), supporting the notion that docking and priming are closely related.

Overall, these results demonstrate the critical importance of the Munc13-1 $C_2$C domain for synaptic vesicle docking, priming and, particularly, Ca$^{2+}$-triggered neurotransmitter release. Moreover, the correlation between the physiological effects caused by the mutations and those caused on liposome clustering and membrane fusion provide strong evidence that the ability of Munc13-1 to bridge membranes is crucial for neurotransmitter release.

## Discussion

Great advances have been recently made in understanding the mechanism of neurotransmitter release, including the fundamental concept that Munc18-1 and Munc13s orchestrate SNARE complex assembly in an NSF-SNAP-resistant manner (*Ma et al., 2013*) that explains at least in part the total abrogation of neurotransmitter release observed in the absence of Munc18-1 or Munc13s (*Richmond et al., 1999*; *Varoqueaux et al., 2002*; *Verhage et al., 2000*). Nevertheless, the actual pathway of SNARE complex assembly is still under intense investigation. The critical role of Munc13s in this process has generally been associated to the activity of its MUN domain in facilitating opening of syntaxin-1 (*Ma et al., 2011*; *Richmond et al., 2001*; *Wang et al., 2017*; *Yang et al., 2015*), but this activity alone does not account for the functional importance of the Munc13 $C_2$C domain, which was suggested by diverse studies (*Liu et al., 2016*; *Madison et al., 2005*; *Stevens et al., 2005*) and is further supported here (*Figure 1*). An attractive model that assigned a critical function to the $C_2$C domain postulated that Munc13-1 can bridge the synaptic vesicle and plasma membranes through interactions involving the $C_2$C domain and the $C_1$-$C_2$B region, respectively (*Figure 1—figure supplement 1*). However, the physiological relevance of this model had not been examined. Here, we provide compelling evidence that the highly conserved C-terminal region of Munc13-1 can indeed bridge two membranes, that the $C_2$C domain is critical for this activity, and that membrane bridging is a key aspect of the function of Munc13-1 in synaptic vesicle docking, priming and fusion. The importance of this bridging function is emphasized by the finding that a single point mutation in this 200 kDa protein abolishes neurotransmitter release almost completely.

This dramatic result suggests that membrane bridging may in fact constitute the primary function of Munc13s, although this notion does not diminish the importance of their role in opening syntaxin-1 because the two activities are likely coupled. Formation of SNARE complexes is hindered not only by the closed conformation of syntaxin-1 (*Dulubova et al., 1999*) but also by the furled conformation of a Munc18-1 loop that prevents synaptobrevin binding and hence hinders the SNARE complex templating function of Munc18-1 (*Sitarska et al., 2017*). The bridge between the synaptic vesicle and plasma membranes provided by Munc13-1 (*Figure 1—figure supplement 1*) is expected to dramatically increase the number of productive encounters between synaptobrevin and the syntaxin-1-

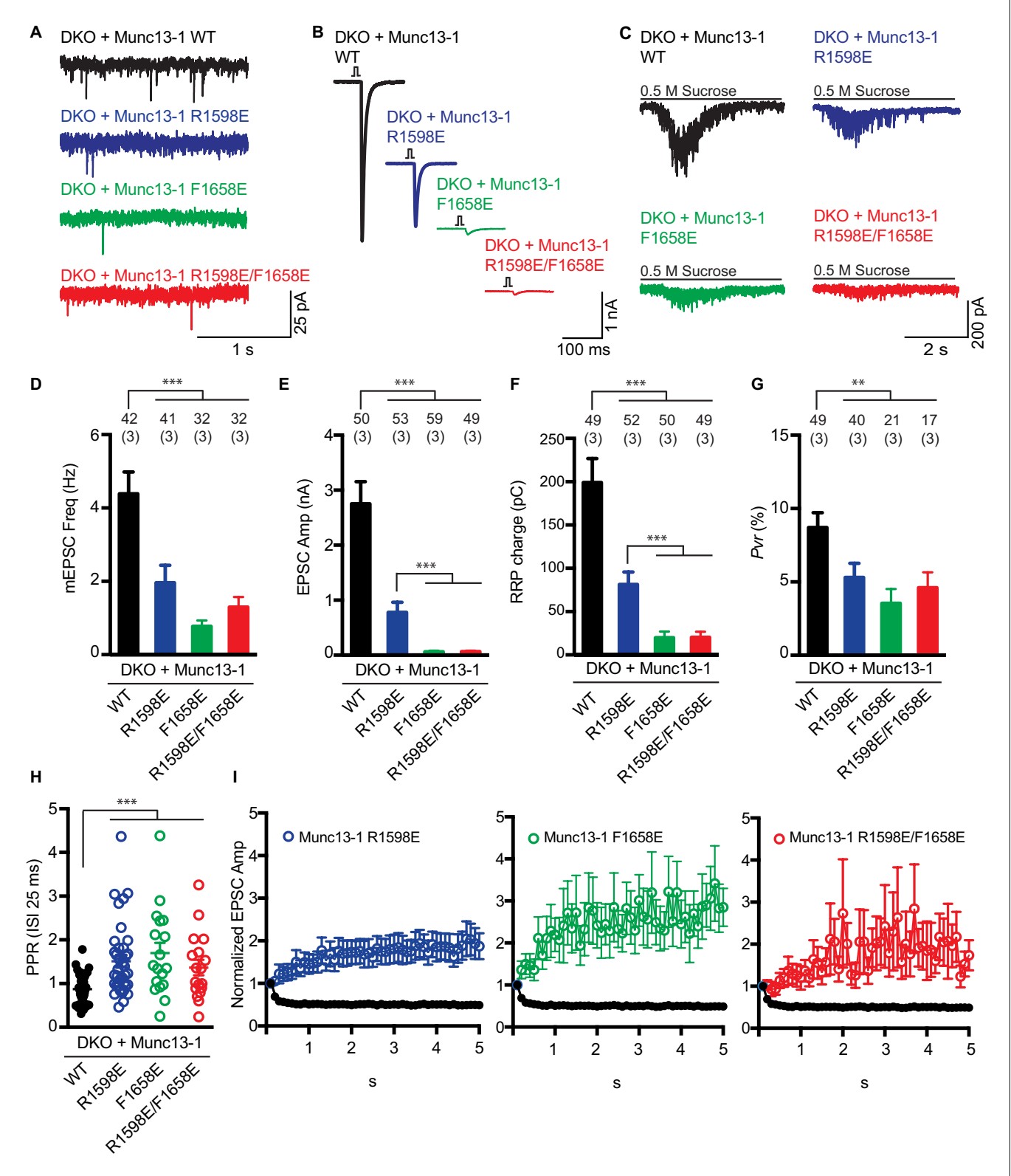

**Figure 7.** Electrophysiological analysis of the functional effects of mutations in the putative membrane-binding loops of the Munc13-1 $C_2C$ domain. (A–C) Representative mEPSCs (A), EPSCs (B), and postsynaptic currents evoked by 0.5 M sucrose (C) in *Munc13-1/2* DKO neurons expressing WT (black), R1598E mutant (blue), F1658E mutant (green) or R1598E/F1658E mutant (red) Munc13-1. (D–G) Mean mEPSC frequencies (D), EPSC amplitudes (E), RRP charges (F) and Pvr (G), measured in the *Munc13-1/2* DKO neurons rescued with WT Munc13-1 and the indicated Munc13-1 mutants. (H) Paired-pulse

*Figure 7 continued on next page*

*Figure 7 continued*

ratios of *Munc13-1/2* DKO neurons rescued with the WT Munc13-1 and the indicated Munc13-1 mutants. (I) Normalized EPSC amplitudes in response to a 10 Hz AP train in *Munc13-1/2* DKO neurons rescued with WT (black), R1598E mutant (blue), F1658E mutant (green) or R1598E/F1658E mutant (red) Munc13-1. Numbers above the bars represent number of neurons pooled together of each group. Numbers in parentheses represent number of cultures or replicates used. All data are mean ±SEM. Significance and p values were determined by Kruskal Wallis test followed by a multiple comparison. *p<0.05, **p<0.01; ***p<0.001.

DOI: https://doi.org/10.7554/eLife.42806.014

The following source data and figure supplements are available for figure 7:

**Source data 1.** Numerical description and statistics of data presented in *Figure 7*.
DOI: https://doi.org/10.7554/eLife.42806.017

**Figure supplement 1.** Localization and expression levels of WT Munc13-1 and Munc13-1 mutants in rescue experiments.
DOI: https://doi.org/10.7554/eLife.42806.015

**Figure supplement 2.** Electrophysiological analysis of the functional effects of overexpressing Munc13-1 bearing mutations in the putative membrane-binding loops and the polybasic region of the $C_2C$ domain.
DOI: https://doi.org/10.7554/eLife.42806.016

Munc18-1 complex to initiate SNARE complex formation (*Xu et al., 2017*), further facilitated by the activity of the Munc13-1 MUN domain in opening syntaxin-1 (*Ma et al., 2011*; *Wang et al., 2017*; *Yang et al., 2015*). Note also that a membrane bridging function for Munc13s is not surprising given the homology of their MUN domain with tethering factors from different membrane compartments (*Pei et al., 2009*; *Yu and Hughson, 2010*). However, these factors normally do not contain $C_1$ or $C_2$ domains. The incorporation of membrane-binding $C_1$ and $C_2$ domains at both ends of the Munc13 MUN domain may have occurred during evolution to provide opportunities for regulation of this membrane-bridging activity, as exquisite regulation is a hallmark of neurotransmitter release and Munc13-1 acts as a master regulator of this process (*Rizo, 2018*). The $C_1$ and $C_2B$ domains of Munc13s are involved in DAG-phorbol ester-dependent potentiation of release (*Basu et al., 2007*; *Rhee et al., 2002*) and $Ca^{2+}$-$PIP_2$-dependent short-term plasticity (*Shin et al., 2010*), respectively. The $C_2C$ domain is not known to be involved in plasticity, but it is tempting to speculate that as yet unidentified mechanisms (e.g. phosphorylation) may modulate $C_2C$ domain activity to regulate neurotransmitter release.

The finding that the $C_2C$ deletion and the mutations in the Munc13-1 $C_2C$ domain described here impair synaptic vesicle docking, priming and release (*Figures 1* and *7–9*), in correlation with the impairments they cause in liposome clustering and fusion in vitro (*Figures 4–6*), suggests that the membrane-bridging activity of the Munc13 C-terminal region is important for more than one of the steps that lead to release. The role in docking-priming is not unexpected, as SNARE complex assembly is believed to be necessary for vesicle docking using the definition that has become widely used recently and we adopt here, that is contact between the vesicle and plasma membranes (*Imig et al., 2014*) (note that, with this definition, docking and priming may constitute the same event, although this equivalence is not fully established [*Rizo, 2018*]). Moreover, the phenotypic spectra in Munc13-1 mutants and in syntaxin-1 titration experiments are highly similar (*Arancillo et al., 2013*), supporting the hypothesis that Munc13-1 membrane bridging and SNARE complex assembly are tightly linked. Note however that, in vivo, synaptic vesicles are believed to be tethered to the active zone through other mechanisms, for instance through RIM-Rab3 interactions, and Munc13 function may be partially redundant with that of CAPS, which contains a MUN domain and membrane-binding domains and also supports SNARE-dependent fusion in reconstitution assays (*James et al., 2009*) (reviewed in *Rizo and Südhof, 2012*). Such redundancy may explain the finding that the effects of the Munc13-1 $C_2C$ domain mutations on the liposome clustering and fusion assays in vitro (*Figures 5–6*) are stronger than those observed in vesicle docking and priming in neurons (*Figures 7–9*).

Interestingly, the Munc13-1 $C_2C$ mutations also disrupt $Ca^{2+}$-triggered neurotransmitter release at least as much as they impair docking and priming. In particular, evoked release is almost abolished by the F1658E and R1598E/F1658E mutations (*Figure 7E*), which correlates with the finding that these mutations abolish liposome clustering and fusion in vitro (*Figures 5* and *6*). These observations suggest that the membrane bridging activity of Munc13-1 is as important for release itself as for vesicle docking-priming. Hence, it seems likely that Munc13-1 still bridges the vesicle and plasma membranes after SNARE complex formation and contributes to controlling the probability of $Ca^{2+}$-

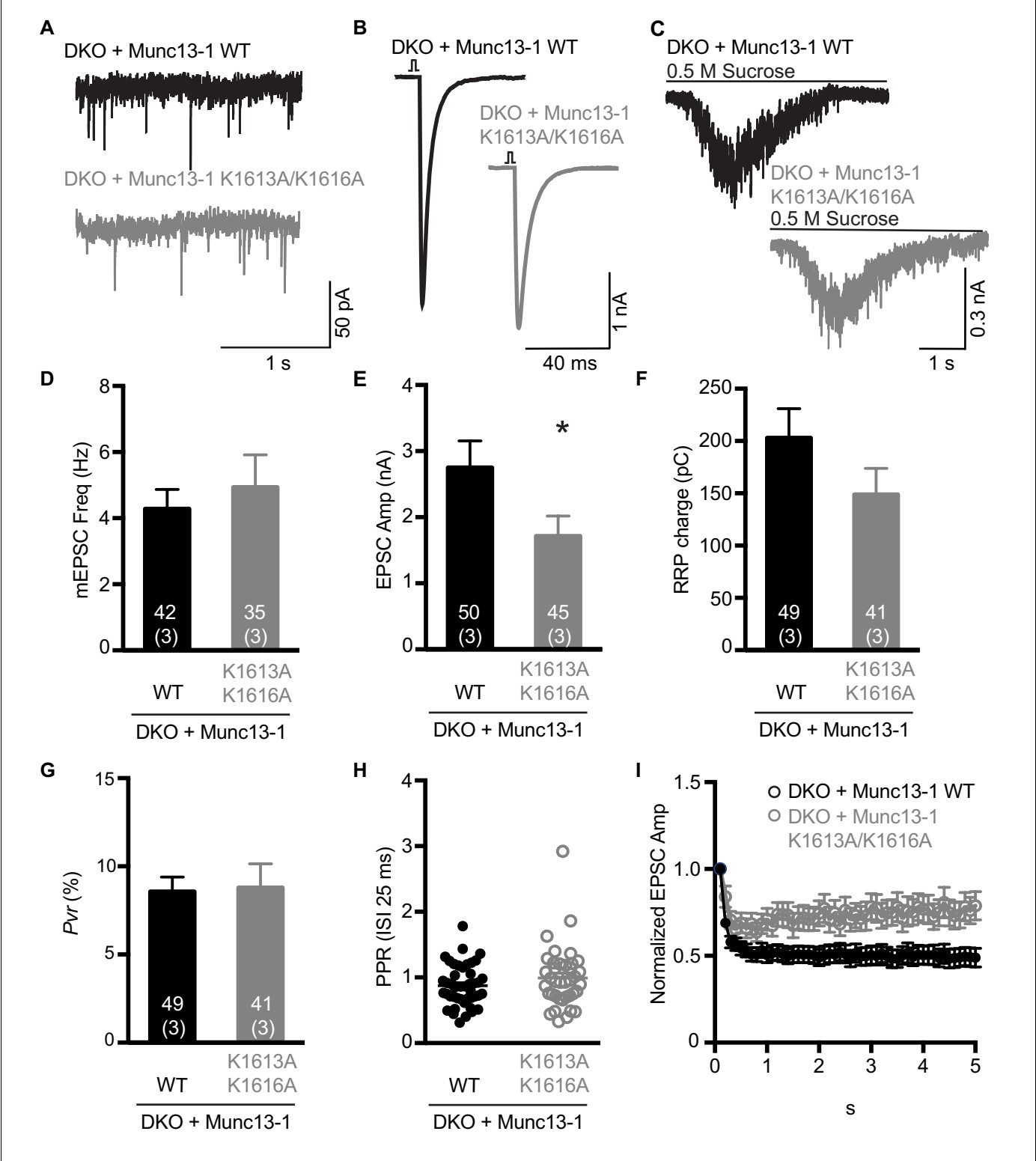

**Figure 8.** Electrophysiological analysis of the functional effects of mutations in the polybasic region of the Munc13-1 $C_2C$ domain. (**A–C**) Examples of mEPSC (**A**), EPSCs (**B**) and sucrose induced currents (**C**) recorded from DKO neurons expressing WT Munc13-1 (black) or a Munc13-1 with a double point mutation at the polybasic stretch within the $C_2C$ domain, Munc13-1 K1613A/K1616A (grey). (**D–F**) Plots showing the average mEPSC frequencies (**D**), EPSC amplitudes (**E**) and RRP charges (**F**), obtained from the DKO neurons rescued with WT or K1613A/K1616A mutant Munc13-1. (**G**) Calculated *Pvr* in % for Munc13-1 WT and for the polybasic mutant K1613A/K1616A. (**H**) Graph showing the average paired-pulse ratios calculated from 2 APs with

*Figure 8 continued on next page*

*Figure 8 continued*

ISI of 25 ms (40 Hz) of DKO rescued with WT or K1613A/K1616A mutant Munc13-1. (**I**) Analysis of EPSC amplitudes in response to a train of 50 AP with an ISI of 100 ms (10 Hz) normalized to the first EPSC and plotted over time for the WT or K1613A/K1616A mutant Munc13-1 rescues. Numbers within the bars represent the number of neurons pooled together of each group. Numbers in parentheses represent the number of cultures or replicates used. All data are mean ± SEM. Significance and p values were determined by Mann Whitney test. *p<0.05.

DOI: https://doi.org/10.7554/eLife.42806.018

The following source data is available for figure 8:

**Source data 1.** Numerical description and statistics of data presented in *Figure 8*.

DOI: https://doi.org/10.7554/eLife.42806.019

triggered synaptic vesicle fusion. It is also plausible that the $C_2C$ domain mutations impair only SNARE complex assembly and a lower number of assembled SNARE complexes results in a stronger impairment of $Ca^{2+}$-triggered fusion than of sucrose-induced release or the number of docked vesicles. Both explanations are not mutually exclusive, but the notion that Munc13-1 still bridges the primed vesicles to the plasma membrane, forming part of the macromolecular assembly that triggers fusion, is attractive because it can explain the multiple and distinct effects of Munc13-1 mutations on vesicular release probability (*Figures 1F* and *7G*, and *Basu et al., 2007*; *Junge et al., 2004*; *Shin et al., 2010*; *Xu et al., 2017*). In particular, the finding that phorbol ester stimulation of Munc13-1 $C_1$ domains acutely increases release probability without changing the number of docked and primed vesicles (*Basu et al., 2007*; *Camacho et al., 2017*) strongly suggests that modulation of the Munc13 bridging function may directly regulate the efficiency of the vesicle fusion reaction. This model is also consistent with recent super-resolution imaging data showing that mammalian Munc13-1 and invertebrate Unc13 form supramolecular assemblies that appear to define the sites for neurotransmitter release in the presynaptic terminal (*Reddy-Alla et al., 2017*; *Sakamoto et al., 2018*).

It is worth noting that the Munc13-1 C-terminal region can likely bridge two membranes in at least two different orientations that can favor SNARE complex formation and/or fusion to different extents, or can also inhibit these events, thus acting as a 'gatekeeper' of release. This notion emerged from the finding that the $C_1$ and $C_2B$ domains have their respective DAG- and $Ca^{2+}$-$PIP_2$-binding sites next to each other and can thus cooperate in binding to the plasma membrane in a defined, slanted orientation, but these domains also form a polybasic region that can bind to membranes in a different orientation, more perpendicular to the membrane (*Xu et al., 2017*). This model provides a basis to understand DAG- and $Ca^{2+}$-dependent presynaptic plasticity that depends on Munc13, and explains the observation that membrane fusion does not occur or is very slow in the absence of $Ca^{2+}$ even though Munc13-1 $C_1C_2BMUNC_2C$ bridges membranes under these conditions (*Figure 3*), whereas fusion is fast upon $Ca^{2+}$-binding to the Munc13-1 $C_2B$ domain (*Liu et al., 2016*) (e.g. *Figures 4B* and *6A*). Mutagenesis studies of *C. elegans* Unc13 support the idea that Unc13 can exist in two states, one that inhibits release and another that activates release and is favored by $Ca^{2+}$-binding to the $C_2B$ domain (*Michelassi et al., 2017*). Our cryo-ET images, which were acquired in the absence of $Ca^{2+}$, show that Munc13-1 $C_1C_2BMUNC_2C$ can bridge two membranes in a range of orientations, some of which would prevent the membranes from coming closer while others could favor initiation of SNARE complex assembly. We previously proposed that $Ca^{2+}$, DAG and $PIP_2$ favor more slanted orientations that can facilitate SNARE complex formation more efficiently and/or membrane fusion (*Xu et al., 2017*). Extensive studies varying these different factors under conditions that prevent membrane fusion will be required to test this proposal.

Further research will also be required to investigate how the membrane-bridging activity of Munc13-1 is coupled to other functions, such as its role in opening syntaxin-1. In this context, a recent report has described the crystal structure of the Munc13-1 MUN bound to a fragment spanning the juxtamembrane region of synaptobrevin and has suggested that this interaction is crucial for Munc13-1 to stimulate the transition from the closed syntaxin-1-Munc18-1 complex to the SNARE complex (*Wang et al., 2019*). The relevance of this structure is unclear, as the synaptobrevin juxtamembrane region contains multiple basic and aromatic residues that render it highly promiscuous and thus able to bind not only to Munc13-1 but also to Munc18-1 (*Xu et al., 2010*) and to phospholipids (*Brewer et al., 2011*), which are natively close to this region. Moreover, mutating two

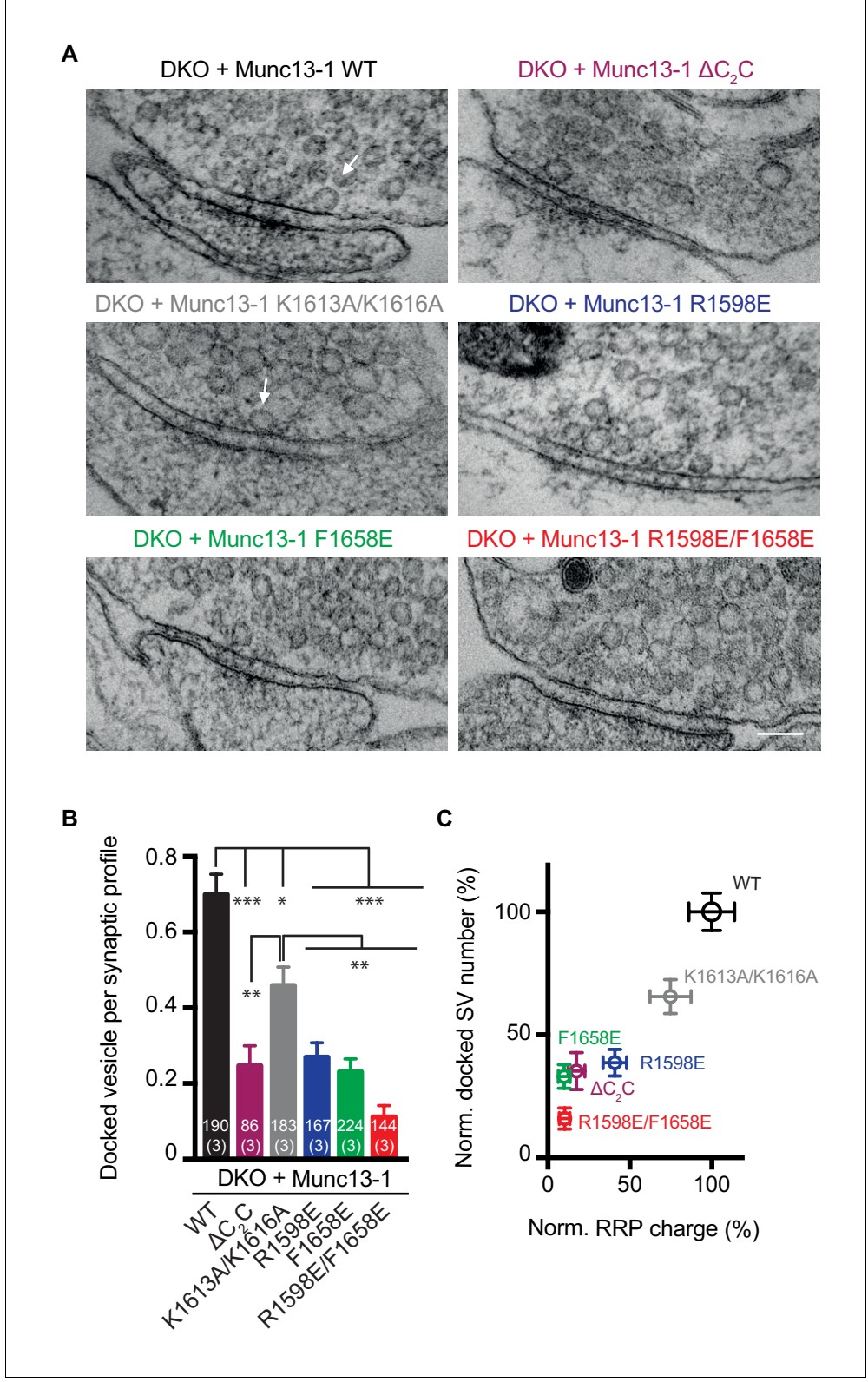

**Figure 9.** Effects of mutations in the Munc13-1 $C_2C$ domain on synaptic vesicle docking. (**A**) Electron micrographs of synapses from DKO hippocampal cultures rescued with WT Munc13-1 and the indicated Munc13-1 $C_2C$ mutants. White arrows indicate docked synaptic vesicles making contact with plasma membrane active zones. Scale bar represents 100 nm. (**B**) Mean number of docked synaptic vesicles per synaptic profile for WT Munc13-1

*Figure 9 continued on next page*

*Figure 9 continued*

and the indicated Munc13-1 $C_2C$ mutants. (C) Plot showing the correlation between primed and docked synaptic vesicles measured after the rescues with WT Munc13-1 and the indicated Munc13-1 mutants. Numbers in bars are the number of synapses pooled together for each group. Numbers in parentheses represent the number of cultures or replicates used. Error bars represent s.e.m. Significance and p values were determined by Kruskal Wallis test followed by a multiple comparison. Values indicate mean ± SEM. *$p < 0.05$; **$p < 0.01$; ***$p < 0.001$; ****$p < 0.0001$.

DOI: https://doi.org/10.7554/eLife.42806.020

The following source data is available for figure 9:

**Source data 1.** Numerical description and statistics of data presented in *Figure 9*.
DOI: https://doi.org/10.7554/eLife.42806.021

basic residues of the juxtamembrane region (R86 and K87) abrogated the MUN-synaptobrevin interaction (*Wang et al., 2019*) but did not impair neurotransmitter release (*Maximov et al., 2009*), and the binding mode observed in the crystal structure would likely lead to steric clashes of the C-terminal region of the MUN domain with the vesicle membrane. Nevertheless, we cannot completely rule out that this interaction might cooperate with binding of the $C_2C$ domain to the membrane. Note also that, although our NMR data suggest that the $C_2C$ domain does not contribute to syntaxin-1 binding (*Figure 2—figure supplement 1C–E*), weak interactions of the MUN domain with syntaxin-1 are believed to be critical to overcome the energy barrier to open its conformation (*Ma et al., 2011*; *Wang et al., 2017*). In addition, Munc13 has been shown to increase the fidelity of SNARE complex assembly by decreasing the number of SNARE complexes that are formed in an antiparallel orientation (*Lai et al., 2017*), indicating that there are additional interactions of Munc13 with the SNAREs that are functionally important. Such interactions may be reminiscent of those found between tethering factors homologous to the Munc13 MUN domain and their cognate SNAREs (*Yu and Hughson, 2010*). It is not surprising that a large, highly conserved component of the release machinery such as Munc13-1 has multiple important roles. It appears that much is currently known, but there is still much to learn.

## Materials and methods

### Plasmids and recombinant proteins

Expression and purification of full-length *Homo sapiens* SNAP-25A (with its four cysteines mutated to serine), full-length *Rattus norvegicus* synaptobrevin-2, full-length *Rattus norvegicus* Munc18-1, full-length *Cricetulus griseus* NSF V155M mutant, full-length *Bos taurus* α-SNAP and *Rattus norvegicus* syntaxin-1 (2–253) in *E. coli* were described previously (*Chen et al., 2006*; *Dulubova et al., 2007*; *Dulubova et al., 1999*; *Ma et al., 2013*). All recombinant *Rattus norvegicus* Munc13-1 fragments contained a deletion in a large variable loop (residues 1408–1452) that improves the solubility (*Ma et al., 2011*). Expression and purification of Munc13-1 $C_1C_2$BMUNC$_2$C (residues 529–1735, Δ1408–1452) in Sf9 cells was described earlier (*Liu et al., 2016*). Standard PCR-based recombinant DNA techniques with custom-designed primers were used to derive vectors to express other Munc13-1 fragments, including vectors to express Munc13-1 $C_1C_2$BMUN1516 (residues 529–1516, Δ1408–1452) in Sf9 insect cells and in *E. coli*, and vectors to express the MUN domain (residues 859–1516, Δ1408–1452), MUNC$_2$C (residues 859–1735, Δ1408–1452) and $C_1C_2$BMUNC$_2$C (residues 529–1735, Δ1408–1452) (WT and K1613A/K1616A, R1598E, F1658E and R1598E/F1658E mutants) in *E. coli*. The constructs to express the Munc13-1 $C_1C_2$BMUN1516 and $C_1C_2$BMUNC$_2$C in *E. coli* were prepared by copying the corresponding Munc13-1 sequences from the vector used for Sf9 cell expression into a pET28a vector kindly provided by Reinhard Jahn (*Kreutzberger et al., 2017*). Expression and purification of $C_1C_2$BMUN1516 in Sf9 insect cells was performed as described earlier for the $C_1C_2$BMUNC$_2$C fragment (*Liu et al., 2016*). Expression and purification of the MUN domain and the MUNC$_2$C fragment in *E. coli* was performed as described previously for a slightly longer fragment spanning the MUN domain (residues 859–1531, Δ1408–1452) (*Ma et al., 2011*). Uniform $^{15}$N-labeling and $^2$H,$^{13}$CH$_3$-ILV-labeling were accomplished as described previously (*Dulubova et al., 1999*; *Tugarinov et al., 2004*).

Expression and purification of $His_6$-Munc13-1 $C_1C_2BMUN1516$ and $C_1C_2BMUNC_2C$ (WT and mutants) encoded in a pET28a vector was performed in *E. coli* BL21 (DE3) cells. Transformed cells were grown in the presence of 50 µg/ml kanamycin to an $OD_{600}$ of ~0.8 and induced overnight at 16°C with 500 µM IPTG. Cells were harvested by centrifugation and re-suspended in 50 mM Tris, pH 8, 250 mM NaCl, 1 mM TCEP, 10% glycerol (v/v) prior to lysis. Cell lysates were centrifuged for 30 min at 48,000 x g to clarify the lysate and then incubated with Ni-NTA resin for 30 min at room temperature. The resin was washed with re-suspension buffer and re-suspension buffer with 750 mM NaCl to remove contaminants. Nuclease treatment was performed on the beads for 1 hr at room temperature using 250 U of Pierce Universal Nuclease (Thermo Fisher Scientific) per liter of cells. Protein was eluted using re-suspension buffer with 500 mM imidazole and dialyzed against 50 mM Tris, pH 8, 250 mM NaCl, 1 mM TCEP, 2.5 mM $CaCl_2$, 10% glycerol (v/v), overnight at 4°C in the presence of thrombin. The solution was re-applied to Ni-NTA resin to remove any uncleaved protein and diluted twentyfold with 20 mM Tris, pH 8, 1 mM TCEP, 10% glycerol (v/v). Diluted protein was subjected to anion exchange chromatography using a HiTrapQ HP column (GE Life Sciences) and eluted in 20 mM Tris, pH 8, 1 mM TCEP, 10% glycerol (v/v) with a linear gradient from 1% to 50% of 1 M NaCl.

$His_6$-full-length syntaxin-1A encoded in a pET28a was expressed in BL21 (DE3) *E. coli*. Transformed cells were grown in the presence of 50 µg/ml kanamycin to an $OD_{600}$ of ~0.8 and induced overnight at 20°C with 400 µM IPTG. Cells were harvested by centrifugation and re-suspended in extraction buffer (20 mM Hepes, pH 7.4, 500 mM NaCl, 8 mM imidazole) prior to lysis. Cell lysates were centrifuged at 14,500 x g for 20 min. The supernatant was discarded and the pellet containing the protein was re-suspended and pelleted again. The pellet was re-suspended in extraction buffer with 2% Triton-X 100 (Sigma-Aldrich) and 6 M urea and incubated for 1 hr at 4°C to solubilize the protein. Cell debris was pelleted by centrifugation at 48,000 x g for 1 hr and the supernatant was applied to Ni-NTA resin. The resin was washed sequentially with wash buffer (20 mM Hepes, pH 7.4, 500 mM NaCl, 20 mM imidazole) containing 6 M urea, 10% glycerol (v/v), 1% Triton-X 100 and then 20% glycerol (v/v), 1% Triton-X 100 and then 1% Triton-X 100 and finally 0.1% n-Dodecylphosphocholine (DPC; Anatrace). The protein was eluted in elution buffer (20 mM Hepes, pH 7.4, 500 mM NaCl, 400 mM imidazole, 0.1 DPC) and the $His_6$-tag was removed by thrombin cleavage overnight at 4°C. Gel filtration was performed on a Superdex 200 10/300 GL column (GE Life Science) in 20 mM Tris, pH 7.4, 150 mM NaCl, 1 mM TCEP, 0.2% DPC.

## NMR spectroscopy

NMR spectra were acquired at 25°C on Agilent DD2 spectrometers operating at 600 or 800 MHz and equipped with cold probes. $^1$H-$^{13}$C HMQC spectra (*Tugarinov et al., 2004*) were obtained on samples containing 10–15 µM $^2$H,$^{13}$CH$_3$-ILV-labeled Munc13-1 MUN or MUNC$_2$C dissolved in 20 mM Tris (pH 8.0), 150 mM NaCl, 2 mM TCEP, using $D_2O$ as the solvent. $^1$H-$^{15}$N TROSY HSQC spectra (*Zhang et al., 1994*) were obtained with samples containing 30 µM $^{15}$N-labeled syntaxin-1 (2–253) alone or with 30 µM Munc13-1 MUN or MUNC$_2$C in 20 mM Tris, pH 7.4, 125 mM NaCl, 2 mM TCEP, 6% $D_2O$. Total acquisition times were 10 hr and 6.6 hr for $^1$H-$^{13}$C HMQC and $^1$H-$^{15}$N HSQC spectra, respectively. All NMR data were processed using NMRPipe (*Delaglio et al., 1995*) and analyzed with NMR View (*Johnson and Blevins, 1994*).

## Analysis of liposome binding using GST pulldown assays

WT or R1598E/F1658E mutant GST-MUNC$_2$C were expressed in *E. coli* BL21 (DE3) cells and cells were lysed as described above. Assuming an expression yield of 10 mg of GST-fusion protein per liter of cells, an estimated 0.3 mg or 1.2 mg of protein was applied to 150 µL of Glutathione Sepharose 4B resin (GE Life Sciences) and incubated for 1 hr at room temperature. The beads were washed three times with wash buffer (20 mM Tris, pH 7.4, 125 mM NaCl) and then DNA/RNA was removed by nuclease treatment for 1 hr at room temperature. The resin was applied to a Micro Bio-Spin Column (Bio-Rad Laboratories) and washed two more times with wash buffer. The buffer was removed by centrifugation at 200 x g and then 200 µL of rho-liposomes containing 39% POPC, 19% DOPS, 20% POPE, 20% cholesterol, and 2% Rhodamine-PE (Avanti Polar Lipids) (0.125 mM total lipid) were applied to the column for 30 min at room temperature. The soluble portion was eluted by

centrifugation at 200 x g and fluorescence spectra were acquired on a PTI Quantamaster 400 spectrofluorometer with excitation at 540 nm and emission from 570 nm to 650 nm.

## Dynamic light scatting

To prepare phospholipid vesicles, 1-palmitoyl-2-oleoyl-sn-glycero-3-phosphocholine (POPC), 1,2-dioleoyl-sn-glycero-3-phospho-L-serine (DOPS), 1-palmitoyl-2-oleoyl-sn-glycero-3-phosphoethanolamine (POPE), L-a-Phosphatidylinositol-4,5-bisphosphate (PIP$_2$), 1-palmitoyl-2-oleoyl-$sn$-glycerol (DAG), and cholesterol dissolved in chloroform were mixed at the desired ratios and then dried under a stream of nitrogen gas. The dried vesicles were left overnight in a vacuum chamber to remove the organic solvent. The next day the lipid films were hydrated with 25 mM HEPES, pH 7.4, 150 mM KCl, 10% glycerol (v/v) and vortexed for 5 min followed by five freeze-thaw cycles. Large unilamellar vesicles were prepared by extruding the hydrated lipid solution through a 100 nm polycarbonate filter 31 times with an Avanti Mini-Extruder. PM-liposomes contained 38% POPC, 18% DOPS, 20% POPE, 2% PIP2, 2% DAG, and 20% cholesterol, and SV-liposomes contained 39% POPC, 19% DOPS, 22% POPE, and 20% cholesterol. Liposome clustering induced by Munc13 fragments was analyzed using a Wyatt Dynapro Nanostar (Wyatt Technology) dynamic light scattering instrument equipped with a temperature controlled microsampler as previously described (*Liu et al., 2016*). Briefly, the specified Munc13-1 fragment (500 nM) was incubated at room temperature for 2 min with PM-liposomes (250 µM total lipid) and SV-liposomes (125 µM total lipid) in 25 mM HEPES, pH 7.4, 150 mM KCl, 100 µM EGTA, 10% glycerol (v/v) prior to measuring the particle size. After the addition of 600 µM Ca$^{2+}$ (to achieve a 500 µM free Ca$^{2+}$ concentration) the sample was incubated for an additional 3 min before measurement.

## Liposome fusion assays

Liposome lipid and content mixing assays were performed basically as previously described (*Liu et al., 2016*; *Liu et al., 2017*). To prepare the phospholipid vesicles, POPC, DOPS, POPE, PIP2, DAG, 1,2-dipalmitoyl-$sn$-glycero-3-phosphoethanolamine-N-(7-nitro-2–1,3-benzoxadiazol-4-yl) (ammonium salt) (NBD-PE), 1,2-Dihexadecanoyl-$sn$-glycero-3-phosphoethanolamine (Marina Blue DHPE), and cholesterol in chloroform were mixed at the desired ratio and dried under a stream of nitrogen gas. T-liposomes contained 38% POPC, 18% DOPS, 20% POPE, 2% PIP2, 2% DAG, and 20% cholesterol, and V-liposomes contained 39% POPC, 19% DOPS, 19% POPE, 20% Cholesterol, 1.5% NBD PE, and 1.5% Marina Blue DHPE. The dried lipids were left overnight in a vacuum chamber to remove the organic solvent. The next day the lipid films were hydrated with 25 mM Hepes, pH 7.4, 150 mM KCl, 1 mM TCEP, 2% n-Octyl-β-D-glucoside (β-OG) and 10% glycerol (v/v) by vortexing for 5 min. Rehydrated lipids for T-liposomes were mixed with protein and dye to get a final concentration of 4 mM lipid, 5 µM full-length syntaxin-1, 25 µM full-length SNAP-25, and 4 µM R-phycoerythrin biotin-XX conjugate (Invitrogen). Rehydrated lipids for V-liposomes were mixed with protein and dye to get a final concentration of 4 mM lipid, 8 µM full-length synaptobrevin, and 8 µM Cy5-streptavidin conjugate (Seracare Life Sciences Inc). Lipid mixtures were dialyzed 1 hr, 2 hr and overnight at 4°C in 25 mM Hepes, pH 7.4, 150 mM KCl, 1 mM TCEP, 10% glycerol (v/v) in the presence of Amberlyte XAD-2 beads (Sigma-Aldrich) to remove the detergent and promote the formation of proteoliposomes. The next day the proteoliposomes were harvested and mixed with Histodenz (Sigma-Aldrich) to a final concentration of 35%. Proteoliposome mixtures were added to a centrifuge tube with 25% Histodenz and 25 mM Hepes, pH 7.4, 150 mM KCl, 1 mM TCEP, 10% glycerol layered on top. The proteoliposomes were spun at 4°C for 1.5 hr at 55,000 RPM in an SW-60 TI rotor and the top layer was collected. Concentrations of the final T-proteoliposomes were measured by the Stewart method (*Stewart, 1980*). V-proteoliposome concentrations were estimated from the UV-vis absorption using a standard curve made using known quantities of liposomes containing 1.5% NBD-PE.

To perform the fusion assays, T-liposomes (250 µM total lipid) were first incubated with 1 µM Munc18, 0.8 µM NSF, 2 µM αSNAP, 2 mM ATP, 2.5 mM Mg$^{2+}$, 5 µM streptavidin, and 100 µM EGTA for 15–25 min at 37°C, and then were mixed with V-liposomes (125 µM total lipid), 1 µM SNAP-25, and wild type Munc13-1 fragments at the specified concentration. After 5 min 0.6 mM Ca$^{2+}$ was added to stimulate fusion, and 1% β-OG was added after 25 min to solubilize the liposomes. The fluorescence signals from Marina Blue (excitation at 370 nm, emission at 465 nm) and

Cy5 (excitation at 565 nm, emission at 670 nm) were recorded on a PTI Quantamaster 400 spectro-fluorometer to monitor lipid and content mixing, respectively. The lipid mixing data were normalized to the maximum fluorescence signal observed upon detergent addition. The content mixing data were normalized to the maximum Cy5 fluorescence observed after detergent addition in control experiments without external streptavidin.

## Liposome co-sedimentation assays

Liposome co-sedimentation assays were performed as described with some modifications (*Shin et al., 2010*). Briefly, lipid mixtures containing 38% POPC, 18% DOPS, 19% POPE, 2% PIP2, 2% DAG, 20% cholesterol, and 1% Rhodamine-PE were dried under a stream of nitrogen gas and kept under vacuum overnight. The next day the lipid film was re-suspended in buffer (25 mM Hepes, pH 7.4, 150 mM KCl, 1 mM TCEP, 500 mM sucrose), frozen and thawed five times, and then extruded through a 100 nm polycarbonate filter 31 times. Liposomes were diluted in sucrose-free buffer and spun at 160,000 x g for 30 min to pellet heavy liposomes. The supernatant was removed and liposomes were re-suspended in sucrose-free buffer. Liposomes were then pelleted at 17,000 x g and re-suspended in sucrose free buffer two more times. The final liposome concentration was estimated based on the absorbance of Rhodamine-PE in a known liposome sample. Liposome solutions containing 2 mM liposome and 2 µM protein were incubated for 30 min at room temperature. The liposomes and bound protein were pelleted by centrifugation at 17,000 x g for 20 min. The supernatant was removed and the liposomes were re-suspended in buffer. Re-suspended samples were boiled for 5 min and analyzed by SDS-PAGE and coomassie blue staining.

## Cryo-Electron tomography

Specimens were prepared following our standard protocol for lipid and content mixing assays (see above), mixing V-liposomes with T-liposomes that had been pre-incubated with Munc18-1, NSF and αSNAP in the presence of Munc13-1 $C_1C_2BMUNC_2C$ fragment and 0.1 µM EGTA. 3 µL of the solution were added to a Lacey carbon grid (200-mesh; Electron Microscopy Sciences) that was negatively glow-discharged for 30 s at 30 mA. 1 µL of 10-fold concentrated solution of 10 nm BSA colloidal gold (Sigma-Aldrich, St. Louis, MO) was quickly mixed into the vesicle solution (*Iancu et al., 2006*), before blotting excess liquid away for ~1.5 s using Whatman filter paper and plunge-freezing the grid in liquid ethane using a CP3 plunge-freezing machine (Gatan, Pleasanton CA). The process from mixing V- and T-liposomes to cryo-immobilization took ~40 s.

The vitrified vesicle samples were imaged using a Titan Krios 300 kV transmission electron microscope (FEI, Hillsboro, OR) equipped with a post-column Gatan imaging filter (Gatan, Pleasanton CA), and a Volta Phase Plate (FEI). The SerialEM software was used to collect tilt series under low-dose mode (*Mastronarde, 2005*). Tilt series were recorded using a dose-symmetric tilting scheme (*Hagen et al., 2017*) and a tilting range from −60° to +60° with an increment of 2°. Images were recorded at 26,000 magnification on a K2 Summit direct electron detector (Gatan, Pleasanton CA) with an effective pixel size of 5.5 Å, and 16 frames were recorded over 5.6 s exposure at a dose rate of 7.8 electrons/pixel/s for each tilt image. The cumulative dose was ~100 $e^-/Å^2$ per tilt series. The defocus was set to −0.5 µm (with phase plate) and the energy filter was in zero-loss mode with a slit width of 20 eV. The tilt series images were aligned and reconstructed in the IMOD software package (*Kremer et al., 1996*) using fiducial alignment and weighted back-projection. To reduce noise, the cryo-tomograms were either binned or slightly filtered using a weighted median filter. For 3D representation, selected areas of the cryo-tomograms were graphically modeled using the modeling tools in IMOD.

## Homology modeling

The SWISS-MODEL server (*Waterhouse et al., 2018*) was used to perform homology modeling of the C-terminal sequence spanning the Munc13-1 $C_2C$ domain (residues 1532–1735). Templates for model building were selected based on the Global Model Quality Estimate (GMQE) score and sequence identity. Final models were built using the RIM1 $C_2B$ domain, synaptotagmin-1 $C_2B$ domain, synaptotagmin-3 $C_2A$ domain and PKC gamma type $C_2$ domain as templates (PDB accession codes 2Q3X, IUOV, 1DQV and 2UZP, respectively).

## Munc13-1 rescue vectors and lentivirus production

Construction of Munc13-1 full length (WT), truncated Munc13-1 $C_2C$ domain (Munc13-1 $\Delta C_2C$), Munc13-1 R1598E, Munc13-1 F1658E, Munc13-1 R1598E/F1658E and Munc13-1 K1613A/K1616A constructs was performed by PCR amplification from rat *Unc13a* splice variant (*Basu et al., 2005*). All PCR products were generated with the appropriate pairs of forward primer and reverse primer harboring a 3xFLAG sequence (Sigma-Aldrich, Hamburg, Germany). The corresponding PCR products with the flag sequence were fused to a P2A linker (*Kim et al., 2011*) after a nuclear localized GFP sequence. All Munc13-1-flag bicistronic constructs were subsequently cloned into a lentiviral shuttle vector under the expressional regulation of human synapsin-1 promoter. Lentiviral particles were produced and concentrated as described previously (*Lois et al., 2002*).

## Hippocampal neuronal culture and lentiviral infection

All animal experiments were conducted according to the rules of the Berlin state government agency for Health and Social Services and the animal welfare committees of Charité Medical University Berlin, Germany (license no. T 0220/09). Primary neuronal hippocampal cultures were prepared from embryonic day 18.5 *Munc13-1/2* DKO mouse or postnatal day 0 C57BL/6N mouse (note that the approved mouse gene symbols for *Munc13-1* and *Munc13-2* are *Unc13a* and *Unc13b*, respectively). Hippocampi were dissected and enzymatically treated with 25 units ml$^{-1}$ of papain for 45 min at 37°C. After papain inactivation, hippocampi were mechanically dissociated in Neurobasal-A medium containing B-27, Glutamax and penicillin/streptomycin. Hippocampal neurons were seeded at $3 \times 10^3$ cells onto 30 mm coverslips previously covered with a dotted pattern of microislands of astrocytes for electrophysiological recordings in autaptic cultures, at $100 \times 10^3$ cells onto 6 mm sapphire disks previously covered with the astrocyte feeder layer for high pressure freezing fixation and at a density of $25 \times 10^3$ cells onto 10 mm coverslips previously covered with an astrocyte feeder layer for immunocytochemical staining. 24 hr after plating neurons were infected with the different lentiviral rescue constructs and incubated at 37°C and 5% $CO_2$ for 14–18 days.

## Immunocytochemistry

*Munc13-1/2* DKO or DKO hippocampal neurons infected with the different rescue constructs were fixed in 4% paraformaldehyde in PBS at DIV 16. After fixation neurons were permeabilized in PBS-Tween 20 (PBS-T), quenched in PBS-T containing glycine, blocked in PBS-T containing 5% normal donkey-serum and incubated overnight at 4°C with mouse monoclonal antibody against Flag M2 (1:100; Sigma, F3165) and guinea pig polyclonal antibody VGLUT 1 (1:4000; Synaptic System, 135304). Primary antibodies were labeled with Alexa Fluor 488 Affinipure donkey anti-rabbit IgG and Alexa Fluor 647 Affinipure donkey anti-guinea pig IgG (1:500 each; Jackson ImmunoResearch). Coverslips with the hippocampal cultures were mounted with Mowiol 4–88 antifade medium (Polysciences Europe). Neuronal images were acquired using a Leica TCS SP8 confocal laser-scanning microscope equipped with a 63x oil immersion objective and Leica Application Suite X (LAsX) software. Confocal fluorescent images were taken at $1024 \times 1024$ pixels with a z step size of 0.3 μm. Ten independent neurons for each culture and two different cultures per group were imaged and analyzed using ImageJ software.

## Electrophysiology

Whole-cell voltage clamp recordings were performed on autaptic hippocampal neurons at DIV14-18 at room temperature. Currents were acquired using a Multiclamp 700B amplifier and a Digidata 1440A digitizer (Axon instrument). Series resistance was set at 70% and only cells with series resistances < 10 MΩ were selected. Data were recorded using Clampex 10 software (Axon instrument) at 10 kHz and filtered at 3 kHz. Borosilicate glass pipettes with a resistance around 3 MΩ were used and filled with an intracellular solution contained the following (in mM): 136 KCl, 17.8 HEPES, 1 EGTA, 4.6 MgCl$_2$, 4 Na$_2$ATP, 0.3 Na$_2$GTP, 12 creatine phosphate, and 50 Uml$^{-1}$ phosphocreatine kinase; 300 mOsm; pH 7.4. Neurons were continuously perfused with standard extracellular solution including the following (in mM): 140 NaCl, 2.4 KCl, 10 HEPES, 10 glucose, 2 CaCl$_2$, 4 MgCl$_2$; 300 mOsm; pH 7.4. Spontaneous release was measured by recording mEPSC for 30 s at −70 mV and for an equal amount of time in 3 mM of the glutamate antagonist Kynurenic Acid to detect false positives events. For each cell, data were filtered at 1 kHz and analyzed using template-based miniature

event detection algorithms implemented in the AxoGraph X software. Action potential-evoked EPSCs were elicited by 2 ms somatic depolarization from −70 to 0 mV. To estimate the readily-releasable pool (RRP) size, 500 mM hypertonic sucrose added to standard extracellular solution, was applied for 5 s using a fast-flow system (*Varoqueaux et al., 2002*). For vesicular release probability ($P_{vr}$) calculations, the ratio of EPSC charge to RRP charge was determined. Short term plasticity was examined either by evoking 2 AP with 25 ms interval (40 Hz) or a train of 50 AP at an interval of 100 ms (10 Hz). Data were analyzed offline using Axograph X (Axograph Scientific).

## High-pressure freezing fixation and transmission electron microscopy (TEM)

Hippocampal *Munc13-1/2* DKO neurons expressing the different Munc13-1 WT and $C_2C$ mutants, immersed in the recording solution containing 2 mM $Ca^{2+}$ and 4 mM $Mg^{2+}$, were frozen using the high-pressure freezer EM ICE (Leica). After the cryofixation, samples were processed as previously described (*Watanabe et al., 2013*). Briefly, samples were transferred to an anhydrous acetone solution containing 1% osmium tetroxide, 1% glutaraldehyde and 1% $ddH_20$ and processed for the freeze-substitution. The freeze-substitution was performed in AFS2 (Leica) over a period of two days with the following program: −90°C for 5 hr, 5°C per hour to −20°C, 12 hr at −20°C and 10°C per hour to 20°C. Once at room temperature, samples were en bloc stained with 0.1% uranyl acetate and infiltrated in increasing concentration of a mixture of epoxy resin (Epon 812) and araldite. Subsequently, samples were flat embedded in resin and cured for 48 hr at 60°C. Serial 40–50 nm thick sections were cut using an Ultracut UCT ultramicrotome (Leica) equipped with a diamond knife (Diatome Ultra 45) and collected onto formvar-coated copper grids. Sections were stained with 1% uranyl acetate and lead citrate for ultrastructural examination. The ultrastructure of the synapse was observed using a FEI Tecnai G20 transmission electron microscope (TEM) operated at 80–120 keV and digital images were acquired with a Veleta 2 K × 2 K CCD camera (Olympus) at 35,000x magnification. Synapses were defined as boutons that contains synaptic vesicles attached to a postsynaptic terminal with a visible postsynaptic density. Around 100–200 synaptic profiles per group were collected blindly and numbers of docked synaptic vesicles per active zone were analyzed using a custom-written analysis program developed for ImageJ and Matlab scripts (*Watanabe et al., 2013*).

## Statistics

Electrophysiological and electron microscopy data were acquired and analyzed blinded. To minimize variability among the electrophysiological datasets, an approximately equal number of autaptic neurons were recorded from control and experimental groups per day. Data were collected from 2 to 3 independent hippocampal cultures and after their analysis the WT control group from each independent culture was tested for normality and for statistical significant difference. No significant differences between the 2–3 independent cultures were observed between the WT groups; Kruskal Wallis test p>0.999. Therefore, the data from the 2–3 replicates for each group were pooled together. Data are expressed as mean ± standard error of the mean (SEM). Statistical comparison was performed by Mann Whitney test (in plots with two groups) or by Kruskal-Wallis one-way ANOVA followed by a multiple comparison Dunn's post hoc test (plots with more that two groups). Statistical differences among datasets were considered significant at p<0.05.

## Acknowledgements

We thank, Miriam Petzold, Berit Söhl-Kielczynski, Sabine Lenz, Bettina Brokowski and Katja Pötschke for technical assistance, and the Charité viral core facility for virus production and characterization. Cryo-ET data were collected at the University of Texas Southwestern Medical Center (UTSW) Cryo-Electron Microscopy (Cryo-EM) Facility that is funded in part by the CPRIT Core Facility Support Award RP170644. We thank Dr. Daniel Stoddard for training and maintenance of the UTSW Cryo-EM Facility. Electron micrographs of synapses were acquired at the Electron Microscopy Core Facility Campus Charité Mitte. The Agilent DD2 console of the 800 MHz spectrometer used for the research presented here was purchased with a shared instrumentation grants from the NIH (S10OD018027 to JR). Bradley Quade was supported by NIH Training Grant T32 GM008297. This work was supported by grant I-1304 from the Welch Foundation (to JR), by NIH Research Project

Award R35 NS097333 (to JR), by the Berlin Institute of Health (BIH) Stiftung Charite (to CR) and by the German Research Foundation (DFG) grants SFB958, SFB 1315 and Ro1296/7-1, 8–1 (to CR).

## Additional information

### Funding

| Funder | Grant reference number | Author |
|---|---|---|
| National Institute of General Medical Sciences | T32 GM008297 | Bradley Quade |
| Deutsche Forschungsgemeinschaft | SFB958 | Christian Rosenmund |
| Deutsche Forschungsgemeinschaft | SFB 1315 | Christian Rosenmund |
| Deutsche Forschungsgemeinschaft | Ro1296/7-1 | Christian Rosenmund |
| Deutsche Forschungsgemeinschaft | Ro1296/8-1 | Christian Rosenmund |
| Berlin Institute of Health Stiftung Charite | | Christian Rosenmund |
| National Institute of Neurological Disorders and Stroke | R35 NS097333 | Josep Rizo |
| Welch Foundation | I-1304 | Josep Rizo |
| National Institutes of Health | S10OD018027 | Josep Rizo |

The funders had no role in study design, data collection and interpretation, or the decision to submit the work for publication.

### Author contributions

Bradley Quade, Marcial Camacho, Conceptualization, Formal analysis, Investigation, Methodology, Writing—original draft; Xiaowei Zhao, Formal analysis, Investigation, Methodology, Writing—original draft; Marta Orlando, Thorsten Trimbuch, Formal analysis, Investigation, Methodology, Writing—review and editing; Junjie Xu, Wei Li, Formal analysis, Investigation, Methodology; Daniela Nicastro, Christian Rosenmund, Conceptualization, Formal analysis, Funding acquisition, Investigation, Methodology, Writing—original draft; Josep Rizo, Conceptualization, Formal analysis, Funding acquisition, Investigation, Methodology, Writing—original draft, Project administration

### Author ORCIDs

Marcial Camacho (iD) https://orcid.org/0000-0002-2367-1259
Daniela Nicastro (iD) https://orcid.org/0000-0002-0122-7173
Christian Rosenmund (iD) http://orcid.org/0000-0002-3905-2444
Josep Rizo (iD) http://orcid.org/0000-0003-1773-8311

### Ethics

Animal experimentation: All animal experiments were conducted according to the rules of the Berlin state government agency for Health and Social Services and the animal welfare committees of Charité Medical University Berlin, Germany (license no. T 0220/09).

### Decision letter and Author response

Decision letter https://doi.org/10.7554/eLife.42806.024
Author response https://doi.org/10.7554/eLife.42806.025

## Additional files

### Supplementary files

• Transparent reporting form
DOI: https://doi.org/10.7554/eLife.42806.022

### Data availability

All data generated or analysed during this study are included in the manuscript and supporting files.

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
