## [Decision Letter]

Thank you for submitting your article "Membrane Bridging by Munc13-1 is Crucial for Neurotransmitter Release" for consideration by *eLife*. Your article has been reviewed by Randy Schekman as the Senior Editor and three reviewers, one of whom is a member of our Board of Reviewing Editors. The reviewers have opted to remain anonymous.

The reviewers have discussed the reviews with one another and the Reviewing Editor has drafted this decision to help you prepare a revised submission.

Summary:

In this manuscript, multiple and complementary approaches are being taken to shed light on the function of the C-terminal C2C domain of the protein Munc13, an essential component of the molecular machinery mediating exocytosis of synaptic vesicles. Building on previous work showing that this domain is critical for function, the authors now report that the domain is needed for forming a bridge (cross-link) between synaptic vesicles and the plasma membrane, and that this cross-linking is essential for the function of Munc13.

This study follows up on Liu et al., 2016 by some of the current authors and reaches the same conclusions. However, the data are extended considerably and provide more structural details. The paper therefore is seen as an important contribution to the field that goes significantly beyond the previous state.

Essential revisions:

1) There is no evidence that the C2C domain binds directly to membranes, though the C2 mutations strongly suggest that this is the role of the C2C domain. In previous liposome co-floatation assays (Liu et al., 2016), no liposome binding was detected with the MUN-C2C fragment. Does C2C actually bind liposomes? For instance, the data in Figure 3 may be interpreted differently: that the hydrophobic sequence 1517-1531 is required for clustering irrespective of whether the C2C domain is present or not. Similarly, replacement of arginine 1598 and phenylalanine 1658 with negatively charged glutamate may increase the overall electrostatic repulsion but again does not prove that this region critically contributes to membrane binding. It would strengthen the interpretation considerably if direct binding of C2C can be demonstrated, e.g. by introducing a fluorescent label (or a spin probe) directly into the presumed interaction site, or by another approach showing that these residues bind to lipids.

2) The authors show clustering of vesicles that is attributed to the cross-linking activity of Munc-13. However, the authors performed this experiment in the presence of SNAREs, NSF, and Munc18. According to their model, clustering should be observable independent of these other proteins (including SNAREs). What is the role of these additional proteins – are they required? In particular, are the cross-links observed by cryo-EM formed between any liposomes (protein-free or not), do they form between v-SNARE-only or t-SNARE-only vesicles, or do they specifically cross-link v- and t-SNARE vesicles? We suggest to use cryo EM (or another, less labor-intensive method) to test whether membrane bridging by wt C1C2BMUNC2C is significantly lower with t- or v-liposomes only. Presently, it cannot be ruled out that cross-linking is indirect, i.e. that Munc13 interactions with the SNAREs (or other proteins) mediate clustering. Similarly, it is possible that Munc13 activates these components, and the SNAREs cluster vesicles. Open syntaxin completely bypasses the docking defects of unc-13 (Hammarlund, 2007), and it is therefore possible that the MUN domain indirectly leads to vesicle clustering by acting through the SNARE proteins.

---

## [Author Response]

This study follows up on Liu et al., 2016 by some of the current authors and reaches the same conclusions. However, the data are extended considerably and provide more structural details. The paper therefore is seen as an important contribution to the field that goes significantly beyond the previous state.

Thank you for the nice comments and constructive criticisms.

Essential revisions:1). There is no evidence that the C2C domain binds directly to membranes, though the C2 mutations strongly suggest that this is the role of the C2C domain. In previous liposome co-floatation assays (Liu et al., 2016), no liposome binding was detected with the MUN-C2C fragment. Does C2C actually bind liposomes? For instance, the data in Figure 3 may be interpreted differently: that the hydrophobic sequence 1517-1531 is required for clustering irrespective of whether the C2C domain is present or not. Similarly, replacement of arginine 1598 and phenylalanine 1658 with negatively charged glutamate may increase the overall electrostatic repulsion but again does not prove that this region critically contributes to membrane binding. It would strengthen the interpretation considerably if direct binding of C2C can be demonstrated, e.g. by introducing a fluorescent label (or a spin probe) directly into the presumed interaction site, or by another approach showing that these residues bind to lipids.

We agree that showing insertion of a fluorescent label attached to a loop of the C_2_C domain of C_1_C_2_BMUNC_2_C would directly demonstrate an interaction with the membranes, but specifically attaching such a label is hindered by the fact that C_1_C_2_BMUNC_2_C contains a large number of cysteines. We have in fact devoted extensive efforts for over two years to attach fluorescent labels using a non-native amino acid approach, but unfortunately all chemical reactions performed to attache the labels led to massive precipitation. We also agree with the concern that the MUNC_2_C fragment did not appear to bind to liposomes in co-floatation assays. In the original manuscript we proposed that this result may arise because binding is weak, but the C_2_C domain may contribute to the liposome clustering activity of C_1_C_2_BMUNC_2_C because of cooperativity between multiple C_1_C_2_BMUNC_2_C molecules. In the revised manuscript we have used a GST-pulldown assay to test for binding of liposomes to MUNC_2_C, as the high local concentrations of proteins accumulated on the resins can favor binding of multiple MUNC_2_C molecules to the same liposome. Indeed, we observed almost quantitative liposome binding to WT MUNC_2_C with this assay. Moreover, the R1598E/F1658E mutation completely abolishes such binding. These results are now shown in a new figure (Figure 2) and described in subsection “A Munc13-1 MUNC2C fragment binds to membranes” of the revised manuscript.

We believe that it is unlikely that the 1517-1531 sequence mediates the clustering ability of C_1_C_2_BMUNC_2_C, or the lipid binding to MUNC_2_C that we now show in Figure 2. However, we agree that this is a valid point and hence that the reasons why we believe that this possibility is unlikely should be explained in the manuscript. With this purpose, we have included the following paragraph (subsection “Membrane bridging by C1C2BMUNC2C is crucial for its ability to support liposome fusion”):

“We note that, in principle, the hydrophobic sequence spanning residues 1517 to 1531 might be responsible for lipid binding, and the effects of the mutations in the C_2_C domain could arise from long-range effects due to changes in the overall electrostatic potential that increase the repulsion with the membranes. However, the fact that the C_2_C domain cannot be expressed in soluble form in isolation while soluble MUNC_2_C is readily expressed suggests that the hydrophobic sequence spanning residues 1517 to 1531 is folded and forms part of the interface between the MUN and C_2_C domains in fragments that contain both domains. Moreover, it is unlikely that long-range effects due to changes in overall electrostatic potential can explain the dramatic disruption of liposome binding (Figure 2) and clustering (Figure 5) caused by mutations in the putative membrane-binding loops of the C_2_C domain. In addition, the K1613A/K1616A mutation removes two positive charges and has a moderate effect on clustering, whereas the F1658E mutation has a very strong effect on clustering while introducing only one negative charge. Conversely, the effects of the mutations can be readily rationalized by the accumulated knowledge on membrane binding to C_2_ domains, which predicts that F1658 is a key residue that inserts into the bilayer to mediate binding (Chapman and Davis, 1998; Rhee et al., 2005), while the polybasic region including K1613 and K1616 is expected to contribute to membrane binding but to a more moderate extent (Li et al., 2006).”

2) The authors show clustering of vesicles that is attributed to the cross-linking activity of Munc-13. However, the authors performed this experiment in the presence of SNAREs, NSF, and Munc18. According to their model, clustering should be observable independent of these other proteins (including SNAREs). What is the role of these additional proteins – are they required? In particular, are the cross-links observed by cryo-EM formed between any liposomes (protein-free or not), do they form between v-SNARE-only or t-SNARE-only vesicles, or do they specifically cross-link v- and t-SNARE vesicles? We suggest to use cryo EM (or another, less labor-intensive method) to test whether membrane bridging by wt C1C2BMUNC2C is significantly lower with t- or v-liposomes only. Presently, it cannot be ruled out that cross-linking is indirect, i.e. that Munc13 interactions with the SNAREs (or other proteins) mediate clustering. Similarly, it is possible that Munc13 activates these components, and the SNAREs cluster vesicles. Open syntaxin completely bypasses the docking defects of unc-13 (Hammarlund, 2007), and it is therefore possible that the MUN domain indirectly leads to vesicle clustering by acting through the SNARE proteins.

The cryo-EM experiments were indeed performed in the presence of Munc18-1, NSF and aSNAP, and using V-liposomes (which contain synaptobrevin and a lipid composition similar to that of synaptic vesicles) and T-liposomes (which contain syntaxin-1-SNAP-25 and a lipid composition that mimics the plasma membrane). The goal was to directly visualize membrane-bridging by C_1_C_2_BMUNC_2_C in the same context as the reconstitution experiments that we believe recapitulate basic aspects of synaptic vesicle fusion. The cryo-EM data supported the notion that C_1_C_2_BMUNC_2_C bridges V- and T-liposomes, which we proposed in Liu et al. 2016 based on the finding that C_1_C_2_BMUNC_2_C cannot cluster V-liposomes alone but clusters mixtures of V- and T-liposomes. We did realize however that the bridging activity might be mediated at least in part by interactions of C_1_C_2_BMUNC_2_C with other proteins, particularly the SNAREs. To rule out this possibility and ensure that the bridging activity is mediated by interactions of C_1_C_2_BMUNC_2_C with membranes, the DLS experiments were performed with liposomes that had the same lipid compositions as V- and T-liposomes, but were protein free (experiments shown in Figure 4A and Figure 5, which were Figure 3A and Figure 4B in the original manuscript). Unfortunately, we still described them in the original manuscript as V- and T-liposomes without noting that they did not contain proteins. We are terribly sorry about this mistake, which we have corrected in the revised manuscript by explaining the different nature of the liposomes and using the term SV- and PM-liposomes (for liposomes with lipid compositions that mimic the synaptic vesicle and plasma membranes, respectively). We inserted the following paragraph in subsection “The C2C domain is required for membrane bridging by the Munc13-1 C-terminal region”:

“To ensure that the bridging activity indeed involves interactions of C_1_C_2_BMUNC_2_C with the membranes and does not depend on binding to proteins, we performed clustering assays monitored by DLS using mixtures of protein-free liposomes with the same lipid compositions as V- and T-liposomes (referred to as SV-liposomes and PM-liposomes because these lipid compositions mimic those of synaptic vesicles and the plasma membrane, respectively). The data showed that C_1_C_2_BMUNC_2_C robustly clusters SV- and PM-liposomes in the absence of Ca^2+^ and that Ca^2+^ does not substantially increase this activity (Figure 4A), as observed previously with V- and T-liposome mixtures (Liu et al., 2016). These results demonstrate that membrane bridging involves direct interactions of C_1_C_2_BMUNC_2_C with the two apposed membranes.”

The DLS experiments unambiguously demonstrate that the liposome bridging activity of C_1_C_2_BMUNC_2_C involves interactions with membranes, as expected because of the presence of membrane-binding domains at opposite ends of the elongated MUN domain structure. The functional importance of this bridging activity in WT neurons is not incompatible with the results of Hammarlund et al., 2007, as the open syntaxin mutant facilitates SNARE complex assembly, making the bridging function of Munc13-1 less crucial for docking. Note that gain-of-function mutations in Munc18-1/Unc18 that also facilitate SNARE complex formation can also bypass the requirement for Munc13-1/Unc13 to some extent, which led to the proposal that autoinhibitory interactions within syntaxin-1 and Munc18-1 hinder SNARE complex assembly to render neurotransmitter release strictly dependent on Munc13-1/Unc13, thus enabling the multiple forms of regulation of release that are mediated by Munc13-1/Unc13 (Sitarska et al., 2017; Park et al., 2017).

We do note that a recent study reported a crystal structure of the Munc13-1 MUN domain bound to a fragment spanning the synaptobrevin juxtamembrane region and claimed that this interaction is important for Munc13-1 function (Wang et al., 2019). We have strong doubts that this structure is biologically relevant because the synaptobrevin juxtamembrane region is very promiscuous, and have included the following paragraph in the Discussion sectopn:

“In this context, a recent report described the crystal structure of the Munc13-1 MUN bound to a fragment spanning the juxtamembrane region of synaptobrevin and suggested that this interaction is crucial for Munc13-1 to stimulate the transition from the closed syntaxin-1-Munc18-1 complex to the SNARE complex (Wang et al., 2019). The relevance of this structure is unclear, as the synaptobrevin juxtamembrane region contains multiple basic and aromatic residues that render it highly promiscuous and thus able to bind not only to Munc13-1 but also to Munc18-1 (Xu et al., 2010) and to phospholipids (Brewer et al., 2011), which are natively close to this region. Moreover, mutating two basic residues of the juxtamembrane region (R86 and K87) abrogated the MUN-synaptobrevin interaction (Wang et al., 2019) but did not impair neurotransmitter release (Maximov et al., 2009), and the binding mode observed in the crystal structure would likely lead to steric clashes of the C-terminal region of the MUN domain with the vesicle membrane. Nevertheless, we cannot completely rule out that this interaction might cooperate with binding of the C_2_C domain to the membrane.”